# LAYERSYNC: SELF-ALIGNING INTERMEDIATE LAYERS

**Yasaman Haghighi**\*   **Bastien van Delft**\*   **Mariam Hassan**   **Alexandre Alahi**
Ecole Polytechnique Fédérale de Lausanne (EPFL)
{yasaman.haghighi,bastien.vandelft,mariam.hassan,alexandre.alahi}@epfl.ch

## ABSTRACT

We propose LayerSync, a domain-agnostic approach for improving the generation quality and the training efficiency of diffusion models. Prior studies have highlighted the connection between the quality of generation and the representations learned by diffusion models, showing that external guidance on model intermediate representations accelerates training. We reconceptualize this paradigm by regularizing diffusion models with their own intermediate representations. Building on the observation that representation quality varies across diffusion model layers, we show that the most semantically rich representations can act as an intrinsic guidance for weaker ones, reducing the need for external supervision. Our approach, LayerSync, is a self-sufficient, plug-and-play regularization term with no overhead on diffusion model training and generalizes beyond the visual domain to other modalities. LayerSync requires no pretrained models or additional data. We extensively evaluate the method on image generation and demonstrate its applicability to other domains such as audio, video, and motion generation. We show that it consistently improves the generation quality and the training efficiency. For example, we speed up the training of flow-based transformers by over $8.75\times$ on ImageNet dataset and improve the generation quality by 23.6%. The code is available at https://github.com/vita-epfl/LayerSync.git.

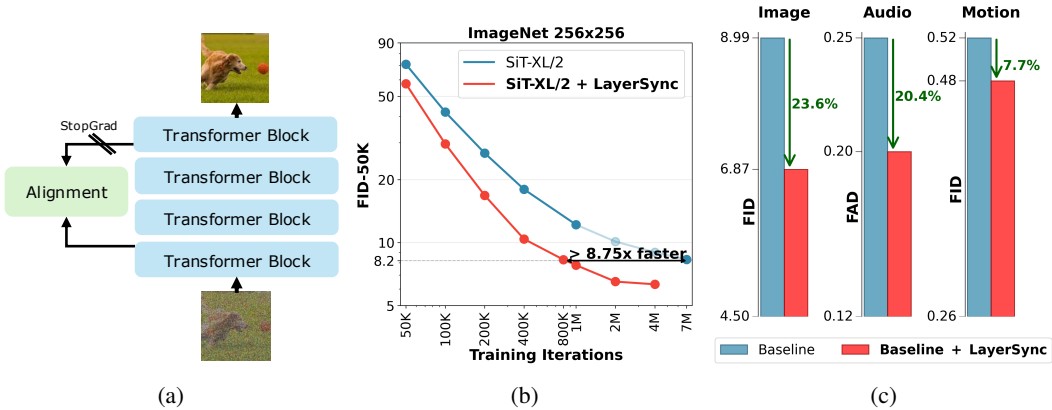

Figure 1: **LayerSync improves training efficiency and generation quality via internal representation alignment.** (a) LayerSync improves representations by aligning shallow layers with semantically rich deep layers. (b) LayerSync achieves over $8.75\times$ training acceleration on the ImageNet 256x256. (c) LayerSync consistently improves generation quality across multiple modalities: 23.6% on FID for images (ImageNet 256×256), 24% on FAD for audio (MTG-Jamendo), and 7.7% on human-motion FID (HumanML3D).

---

\*Equal contribution

# 1 INTRODUCTION

Denoising generative models, such as diffusion (Ho et al., 2020; Song et al., 2020; Song & Ermon, 2019) and flow matching models (Lipman et al., 2022), have demonstrated remarkable success in modeling complex data distributions, achieving state-of-the-art performance across a range of generative tasks. However, this success comes at a significant computation cost. Thus, a new promising line of research has emerged to improve the training efficiency of these models by improving the models' intermediate representations (Yu et al., 2024; Wang et al., 2025; Wang & He, 2025). It has been shown that the quality of a diffusion model's intermediate representations is intrinsically linked to its generative performance. As a result, explicitly guiding these representations accelerates training and improves generation quality (Yu et al., 2024).

Building on this insight, the most dominant approach (Yu et al., 2024; Wang et al., 2025) has been to leverage powerful external guidance from large pre-trained models, by aligning the internal features of a diffusion model with those of high-capacity vision models like DINOv2 (Oquab et al., 2023) or vision-language models (VLMs) like Qwen2-VL (Wang et al., 2024). These methods demonstrate that access to strong semantic features can accelerate training by an order of magnitude. While effective, this paradigm comes with several limitations. It introduces a dependency on massive external models that are themselves costly to train, require large amounts of data, and may not be available for domains beyond natural images. Additionally, this reliance on external data and parameters introduces extra overhead into the training pipeline. For instance, in the case of Wang et al. (2025), training is indeed faster in terms of iterations but involves calling a 9-billion-parameter VLM at each step. These limitations motivate the development of more self-contained and generalizable alternatives.

A recent step in this direction is the Dispersive Loss (Wang & He, 2025), a self-contained regularizer that encourages internal representations to spread out in the feature space, analogous to the repulsive force in contrastive learning (Oord et al., 2018). Although this approach demonstrates the potential for internal regularization, a substantial performance gap remains compared to methods that leverage external representations. In this paper, we propose a self-contained method with a more directed learning signal to reduce this gap.

Our work is motivated by two key observations: First, while diffusion models learn powerful representations, their quality is highly heterogeneous across the model's depth. As demonstrated by previous works (Mukhopadhyay et al., 2024; Kim et al., 2025; Stracke et al., 2025), certain intermediate layers capture more semantically rich and useful information than others. Second, when models incorporate knowledge through external guidance using DINOv2 for instance, regularizing early layers seemed more effective than regularizing the deeper ones (Yu et al., 2024; Wang et al., 2025). Based on these two observations, a clear opportunity presents itself: can the model's own strongest layers act as an intrinsic guidance to improve its weaker ones through self-alignment?

To this end, we propose LayerSync, a simple yet powerful regularization framework that aligns a model's own intermediate layers. LayerSync is a parameter-free, plug-and-play solution that operates without any external models or data, making it a truly self-contained method. LayerSync introduces negligible computational overhead, yet its effectiveness is substantial. Our experiments show that LayerSync consistently outperforms prior self-contained methods across all tested configurations. For image generation, LayerSync accelerates training on ImageNet 256×256 (Deng et al., 2009) by more than 8.75×. This leads to a new state-of-the-art in purely self-supervised image generation on ImageNet, demonstrating the strength of our self-alignment objective and substantially narrowing the gap between self-contained approaches and those relying on external guidance. Furthermore, due to its self-contained nature, LayerSync seamlessly generalizes to other modalities. Our experiments show that for audio generation, LayerSync leads to $21\%$ improvement in FAD-10K on MTG-Jamendo (Bogdanov et al., 2019), to $7.7\%$ improvement in FID for human motion generation on HumanML3D and $54.7\%$ in FVD for video generation on CLEVRER (Yi et al., 2019) (see Appendix A). To the best of our knowledge, it is the first time that a self-contained method proves to accelerate diffusion model training seamlessly across different domains.

Additionally, an analysis of the internal features confirms that LayerSync strengthens the model's representations, leading to a $32.4\%$ improvement in classification and a $63.3\%$ improvement in semantic segmentation accuracy.

Our main contributions are as follows:

- We introduce LayerSync, a minimalist, parameter-free, and self-contained regularization method that leverages a diffusion model's own layers as an intrinsic guidance via self-alignment.
- We demonstrate the domain-agnostic versatility of LayerSync by successfully applying it to image, audio, human motion, and video generation.
- We show that our self-supervised method not only accelerates training but also improves the representations across the model's layers.

## 2 RELATED WORK

**Representation learning with diffusion models.** Denoising Generative Models, including both diffusion (Ho et al., 2020; Song et al., 2020; Song & Ermon, 2019) and flow matching models (Lipman et al., 2022), trained as multi-level denoising autoencoders (Vincent et al., 2008), naturally give rise to discriminative representations. A line of work has specifically evaluated the quality of these representations, showing that diffusion features can be effectively used across a variety of tasks (Mukhopadhyay et al., 2024; Kim et al., 2025; Stracke et al., 2025) and, in some cases, achieve performance comparable to self-supervised representation learning methods (Stracke et al., 2025). However, the quality of the representations varies across model layers, with the final layers, just before the model begins decoding, consistently containing more semantically rich features (Kim et al., 2025; Xiang et al., 2023), regardless of whether the architecture is a U-Net (Ronneberger et al., 2015) or a Transformer (Vaswani et al., 2017). Our work is directly built upon those insights. We demonstrate that the semantically rich representations in the intermediate layers can be leveraged as a guidance signal to enhance the quality of earlier-layer representations.

**Representation regularization for improving diffusion models.** It has been shown that representation quality is closely linked to generative performance (Yu et al., 2024). One line of work improves generation by regularizing model representations through alignment with strong pretrained networks. For example, Yu et al. (2024) aligns diffusion features with self-supervised features from DINOv2 (Oquab et al., 2023), while Wang et al. (2025) demonstrates that leveraging vision–language models (VLMs) (Wang et al., 2024) can yield further improvements. Although such approaches accelerate training and enhance generation quality, they remain constrained by the need for high-quality external representations, which are not readily available in non-visual domains. Additionally, they introduce computational overhead, as pretrained models must be inferred at each training step. Another group of work adopts self-supervised strategies that rely on EMA (Exponential Moving Average) (Tarvainen & Valpola, 2017) models to guide the representations. Zheng et al. (2023) integrates a generative diffusion process with an auxiliary mask reconstruction task. Zhu et al. (2024); Jiang et al. (2025) align representations between teacher and student encoders in a joint embedding space. While being self-contained, such methods increase computational cost by requiring an additional forward pass through the EMA model at each training step. A recent work (Wang & He, 2025) proposes dispersing representations in the feature space, analogous to the repulsive force in contrastive learning. This approach introduces no additional training overhead. Similarly, we present a self-contained, overhead-free solution; however, we leverage semantically richer internal representations to guide and improve the learning of weaker ones.

## 3 METHOD

### 3.1 PRELIMINARIES

We adopt the generalized perspective of stochastic interpolants (Ma et al., 2024), which provides a unifying framework for both flow-based and diffusion-based models. Here is a brief overview; we refer to Appendix Section N for more details.

Stochastic interpolants are generative models that learn to reverse a process that gradually converts a data sample $\mathbf{x}_0$ into simple noise $\epsilon$. This is achieved by defining a path between them:

$$\mathbf{x}_t = \alpha_t \mathbf{x}_0 + \sigma_t \epsilon,$$

where $\alpha_t$ and $\sigma_t$ are functions of time controlling the mix of data and noise at time $t$. To generate new data, the model must learn to travel backward along this path, from noise to data. The direction and speed at any point $\mathbf{x}_t$ and time $t$ is given by a velocity field. The true velocity is the time derivative of the path: $\dot{\alpha}_t\mathbf{x}_0 + \dot{\sigma}_t\epsilon$.

Since this true velocity is unknown during generation, a neural network $v_\theta(\mathbf{x}_t, t)$ is trained to predict it. The model learns by minimizing the velocity loss, which measures the squared difference between the predicted velocity and the ground-truth velocity:

$$\mathcal{L}_{\text{velocity}}(\theta) := \mathbb{E}_{\mathbf{x}_0, \epsilon, t}\left[\|v_\theta(\mathbf{x}_t, t) - (\dot{\alpha}_t\mathbf{x}_0 + \dot{\sigma}_t\epsilon)\|^2\right]. \tag{1}$$

Once trained, the model can generate new data by starting with a random noise sample and following the velocity field it has learned.

## 3.2 DIFFUSION MODELS INTERMEDIATE REPRESENTATIONS

**Representation Hierarchy.** We investigate the representations learned by a pre-trained SiT model (Ma et al., 2024) on ImageNet (Deng et al., 2009). Through linear probing on downstream tasks (classification, segmentation) and Centered Kernel Alignment (CKA) with DINOv2 (Oquab et al., 2023) features, we observe a clear hierarchy in representation quality across layers. As shown in Figure 4, deeper layers exhibit superior discriminative capabilities, consistent with established principles of hierarchical feature learning in deep networks (LeCun et al., 2015). This pattern of increasing semantic richness culminating in a peak before the final decoding blocks is a known characteristic of diffusion model representations (Mukhopadhyay et al., 2024).

**Internal Block Structure of Diffusion Transformers.** Beyond the hierarchy of feature quality, we observe that the internal structure of Diffusion Transformers at convergence follows specific correlation patterns as shown in Figure 2 and discussed in (Raghu et al., 2021; An et al., 2025). These blocks are highly correlated and naturally segregate into three functional groups: (1) an initial group focusing on local features, (2) a middle group of highly correlated blocks capturing global features, and (3) a final group acting as a decoder to project back to the latent space.

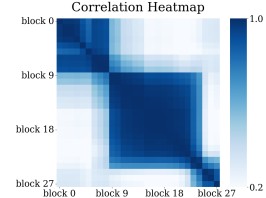

Figure 2: Correlation between the blocks of SiT-XL/2 at Convergence.

## 3.3 LAYERSYNC

We propose a self-contained regularization approach, named **LayerSync**, designed to improve the generation quality and training dynamics of diffusion models. The core principle behind LayerSync is intra-model self-alignment, where the model is trained to guide itself. Building on the hierarchy of representation quality across layers, we use the context-rich deep layers as an "intrinsic guidance" to provide a direct signal to the earlier "weak" layers, thereby enhancing the model's entire feature from within. The extra alignment between the model's intermediate layers aligns with the block structure of the diffusion transformer at convergence.

LayerSync achieves this alignment by maximizing the similarity between the feature representations of designated strong and weak blocks. The similarity is computed for each patch in the representation and then averaged over the whole patch sequence for each image. Let $f_\theta$ be the network transformer and let $f_\theta^k$ designate the network up to the k-th layer. Let $\mathbf{x}_t$ be the input marginal distribution at time $t$, with $t \sim Uniform(0, 1)$ and $x \sim \mathbf{x}_t$, we define the loss for LayerSync between layer $k$ and $k'$ with $k < k'$ as follows:

$$\mathcal{L}_{\text{LayerSync}_{(k,k')}}(\theta) := -\mathbb{E}_{\mathbf{x}_t, t}\left[\frac{1}{N}\sum_{n=1}^{N}\text{sim}\left(f_\theta^k(x)^{[n]}, \text{stopgrad}\left(f_\theta^{k'}(x)^{[n]}\right)\right)\right], \tag{2}$$

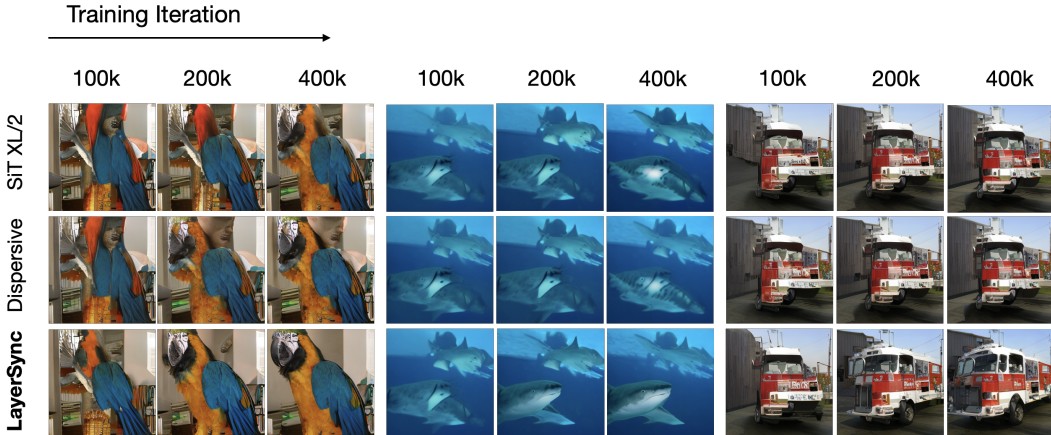

Figure 3: **LayerSync improves generation quality without relying on external representation.** We compare the images generated by SiT-XL/2 when regularized with dispersive and LayerSync. All the models are trained for 400K iterations, share the same noise, sampler, and number of sampling steps, and none of them use classifier-free guidance.

where $n$ is the patch index and $\text{sim}(\cdot, \cdot)$ is a pre-defined similarity function. We experimented with different similarity functions and opted for cosine similarity in all our following experiments. This regularization term is integrated into the primary training objective as a weighted sum:

$$\mathcal{L} := \mathcal{L}_{\text{velocity}} + \lambda\mathcal{L}_{\text{LayerSync}}, \tag{3}$$

where the hyperparameter $\lambda > 0$ balances the standard denoising task with our internal representation alignment. Algorithm 1 in the Appendix summarizes our proposed approach.

While LayerSync internally regularizes the network by aligning representations from its early layers with those from its own deeper, semantically richer layers, beyond the immediate benefit of improving shallow layers, we hypothesize that this method may induce a virtuous cycle in the recursive refinement process. The enhancement of early-layer features is expected to facilitate the learning of more robust deep-layer representations, which subsequently offer a more refined target for internal alignment, potentially leading to a cascading improvement of the model's feature space.

### 3.4 LAYER SELECTION

A key design consideration for LayerSync is the selection of the layers to align. As detailed in Section 3.2, the layers of Diffusion Transformers naturally converge toward a structure in three groups of blocks focusing on local features, global features, and decoding. While our experiments indicate that LayerSync yields performance gains across a wide range of configurations, we observe that respecting this structure leads to optimal acceleration (see Section D).

Based on these observations, we propose a selection strategy guided by three principles. First, drawing on established findings (Xiang et al., 2023) regarding the functional specialization of layers in generative transformers, we exclude the final 20% of blocks from being chosen as the reference layer, as those are primarily specialized for low-level decoding tasks, making them suboptimal as guidance targets. Second, based on the findings from (Raghu et al., 2021), we exclude the very first layers, as prior work finds that early blocks focusing on local features improve performance and generalization (An et al., 2025). Third, to ensure a meaningful semantic gap between the representations, we enforce a minimum distance between the aligned and reference layers (e.g., 8 blocks for SiT-XL and SiT-L and 3 for SiT-B). We validate this heuristic and the robustness of the method to different layer selection through experiments as summarized in Tables 6, 11 and 12.

## 4 EXPERIMENTS

We conduct a comprehensive set of experiments to validate the effectiveness of LayerSync. Our evaluation is structured along three axes:

- We first study extensively the performance and training efficiency of LayerSync in large-scale class-conditional image generation (Section 4.1).

- We then assess the domain-agnostic capabilities of our method by applying it to generative tasks in audio (Section 4.2), human motion (Section 4.3), and video (Appendix Section A).

- Finally, we perform an in-depth analysis to quantify the impact of LayerSync on the quality and structure of the learned internal representations (Section 4.4).

### 4.1 IMAGE GENERATION

**Implementation details.** We strictly follow the setup in SiT (Ma et al., 2024). Specifically, we use ImageNet (Deng et al., 2009) and follow ADM (Dhariwal & Nichol, 2021) for data preprocessing. The processed image will have the resolution of $256 \times 256$ and is then encoded into a compressed vector $z \in \mathbb{R}^{32 \times 32 \times 4}$ using the Stable Diffusion VAE (Rombach et al., 2022). For model configurations, we use the B/2, L/2, and XL/2 architectures by Ma et al. (2024), which process inputs with a patch size of 2. More details about the architectures and the number of parameters are provided in the Section Q. Additional experimental details, including hyperparameter settings and computing resources, are provided in Section L.

**Evaluation metrics.** We report Fréchet inception distance (FID; Heusel et al. (2017)), Inception Score (Salimans et al., 2016), Precision, and Recall (Kynkäänniemi et al., 2019) using 50,000 samples. We provide details of each metric in Section O.

**Baselines.** We compare our results with Dispersive (Wang & He, 2025), the only self-contained, zero-cost method that accelerates training. For the sake of completeness, we also compare our method with several recent diffusion-based generation methods. For pixel-based approaches we compare with ADM (Dhariwal & Nichol, 2021), VDM++ (Kingma & Gao, 2023), Simple diffusion (Hoogeboom et al., 2023), CDM (Ho et al., 2022). For latent-based approaches we compare with LDM (Rombach et al., 2022), U-ViT-H/2 (Bao et al., 2023), DiffiT (Hatamizadeh et al., 2024), MDTv2-XL/2 (Gao et al., 2023), MaskDiT (Zheng et al., 2023), SD-DiT (Zhu et al., 2024), DiT (Peebles & Xie, 2023), and SiT (Ma et al., 2024). We also compare our approach with REPA (Zhang et al., 2025) and REED (Wang et al., 2025), which rely on external representations. Additionally, we compare to two autoregressive methods, VAR (Tian et al., 2024) and D-JEPA (Chen et al., 2024). For both methods, we selected the models with a similar number of parameters to SiT-XL. Finally, we compare our results with the EMA-based method SRA (Jiang et al., 2025). A detailed description of each baseline is provided in Section P.

**Results.** As shown in Table 1, our method consistently improves diffusion transformer training and is more effective than Wang & He (2025). Our method results in $8.75\times$ acceleration compared to SiT-XL baseline, reaching an FID of 8.29 after only 160 epochs, and in $4.7\times$ acceleration compared to the baseline trained with Dispersive Loss. In Table 2 we compare LayerSync with recent state-of-the-art diffusion model approaches. In particular, on SiT-XL/2, we reach FID 1.89 after 800 epochs, setting a new state-of-the-art in pure self-supervised generation, decreasing the gap with methods like Yu et al. (2024) that rely on external representations. We also qualitatively compare the progression of generation results in Figure 3, where we use the same initial noise across different models. Additional comparison metrics with SRA are provided in Table 15.

### 4.2 AUDIO GENERATION

**Implementation details.** We use the MTG-Jamendo dataset (Bogdanov et al., 2019), a large-scale collection containing over 55,000 full-length songs. For training, we process the audio by creating random 10-second samples, which are sampled at a standard rate of 44.1 kHz. We condition using the metadata provided with the dataset by conditioning the generation on the genre and instrument labels associated with each sample. Our audio generation model is an adaptation of the Scalable Interpolant Transformer (SiT-XL) (Ma et al., 2024), consistent with the 28-layer architecture used in

Table 1: FID comparisons of class-conditional generation on ImageNet 256×256. No classifier-free guidance (CFG; Ho & Salimans (2022)) is used. The sampler used is the ODE-based Heun method, except for the last section, which uses the SDE-based Euler method following Ma et al. (2024)

| Model | #Params | Epochs. | FID↓ |
|---|---|---|---|
| SiT-B/2 | 130M | 80 | 36.19 |
| + Dispersive | 130M | 80 | 32.45 (10.3%) |
| + LayerSync | 130M | 80 | **30.00** (17.1%) |
| SiT-L/2 | 458M | 80 | 21.41 |
| + Dispersive | 458M | 80 | 16.68 (22.1%) |
| + LayerSync | 458M | 80 | **14.83** (30.7%) |
| SiT-XL/2 | 675M | 80 | 17.97 |
| + Dispersive | 675M | 80 | 15.95 (11.3%) |
| + LayerSync | 675M | 80 | **11.24** (37.5%) |
| SiT-XL/2 | 675M | 200 | 12.18 |
| + Dispersive | 675M | 200 | 10.64 (12.6%) |
| + LayerSync | 675M | 200 | **8.28** (32.0%) |
| SiT-XL/2 | 675M | 400 | 10.11 |
| + Dispersive | 675M | 400 | 8.81 (12.9%) |
| + LayerSync | 675M | 400 | **6.94** (31.4%) |
| SiT-XL/2 | 675M | 800 | 8.99 |
| + Dispersive | 675M | 800 | 8.08 (10.1%) |
| + LayerSync | 675M | 800 | **6.87** (23.6%) |
| SiT-XL/2 (w/ SDE) | 675M | 1400 | 8.3 |
| + Dispersive | 675M | 800 | 7.71 (7.1%) |
| + Dispersive | 675M | ≥ 1200 | 7.43 (10.5%) |
| + LayerSync | 675M | 160 | **8.29** (0.1%) |
| + LayerSync | 675M | 200 | **7.78** (6.3%) |
| + LayerSync | 675M | 400 | **6.51** (21.6%) |
| + LayerSync | 675M | 800 | **6.32** (23.9%) |

Table 2: System-level comparison on ImageNet 256×256 with CFG. Models with additional CFG scheduling Kynkäänniemi et al. (2024) are marked with an asterisk (*).

| Model | Epochs | FID↓ | IS↑ | Pre.↑ | Rec.↑ |
|---|---|---|---|---|---|
| *Pixel diffusion* | | | | | |
| ADM-U | 400 | 3.94 | 186.7 | 0.82 | 0.52 |
| VDM++ | 560 | 2.40 | 225.3 | – | – |
| Simple diffusion | 800 | 2.77 | 211.8 | – | – |
| CDM | 2160 | 4.88 | 158.7 | – | – |
| *Autoregressive Image Generation* | | | | | |
| VAR-d20 | 350 | 2.57 | 302.6 | 0.83 | 0.56 |
| D-JEPA-L | 480 | 1.58 | 303.1 | 0.80 | 0.61 |
| *Latent diffusion, U-Net* | | | | | |
| LDM-4 | 200 | 3.60 | 247.7 | **0.87** | 0.48 |
| *Latent diffusion, Transformer + U-Net hybrid* | | | | | |
| U-ViT-H/2 | 240 | 2.29 | 263.9 | 0.82 | 0.57 |
| DiffiT* | – | 1.73 | 276.5 | 0.80 | 0.62 |
| MDTv2-XL/2* | 1080 | 1.58 | 314.7 | 0.79 | **0.65** |
| *Latent diffusion, Transformer* | | | | | |
| MaskDiT | 1600 | 2.28 | 276.6 | 0.80 | 0.61 |
| SD-DiT | 480 | 3.23 | - | - | - |
| DiT-XL/2 | 1400 | 2.27 | 278.2 | 0.83 | 0.57 |
| SRA* | 800 | 1.58 | 311.4 | 0.80 | 0.63 |
| SiT-XL/2 | 1400 | 2.06 | 270.3 | 0.82 | 0.59 |
| + REPA | 800 | 1.80 | 284.0 | 0.81 | 0.61 |
| + REPA* | 800 | **1.42** | - | - | - |
| + REED | 200 | 1.80 | 267.5 | 0.81 | 0.61 |
| + Dispersive | ≥ 1200 | 1.97 | - | - | - |
| + LayerSync | 800 | 1.89 | 265.34 | 0.81 | 0.60 |
| + LayerSync* | 800 | 1.49 | 285.2 | 0.80 | 0.63 |

Table 3: Quantitative results for audio generation on the MTG-Jamendo dataset. We report Fréchet Audio Distance (FAD) using CLAP embeddings. Our method significantly outperforms the baseline with no change in parameter count.

| Method | #Params | Epoch | FAD (CLAP) ↓ |
|---|---|---|---|
| SiT-XL (baseline) | 756M | 465 | 0.333 |
| **+ LayerSync (Ours)** | **756M** | **465** | **0.263** (21.0%) |
| SiT-XL (baseline) | 756M | 650 | 0.251 |
| **+ LayerSync (Ours)** | **756M** | **650** | **0.199** (20.7%) |

our vision experiments. The model is configured to operate on patchified latent representations with a patch size of one. These latents are obtained from the pre-trained Variational Autoencoder (VAE) of the Stable Audio Open model (Evans et al., 2025), which provides a compact representation of the raw audio waveforms. The model was trained on 64 GH200 GPUs with a global batch size of 1024. In our experiment, we align layer 8 with 21 using cosine similarity between the patch-wise representations of these two layers.

**Evaluation metrics.** To quantitatively assess the quality and realism of the generated audio, we report the Fréchet Audio Distance (FAD)(Kilgour et al., 2018) with 10,000 samples using the widely-used CLAP embeddings (Zhao et al., 2023).

**Results.** LayerSync improves the final FAD-10K by 20.7% at 650 epochs as seen in Table 3. The model trained with LayerSync reaches the final performance of the baseline model around epoch 500 so 150 epochs earlier. The convergence speed is therefore improved by 23%.

## 4.3 TEXT-CONDITIONAL HUMAN MOTION GENERATION

To demonstrate that LayerSync can be applied in domains with limited datasets and compact architectures, we consider the task of human motion generation. Given a sentence that describes a motion

Table 4: Quantitative results for text-conditional human motion generation task on HumanML3D dataset. LayerSync improves both FID and R-Precision.

| Method | Iter. | FID ↓ | R-Precision ↑ |
|---|---|---|---|
| MDM | 600K | 0.5206 | 0.7202 |
| **+ LayerSync (Ours)** | 600K | **0.4801** (7.7%) | **0.7454** (3.4%) |

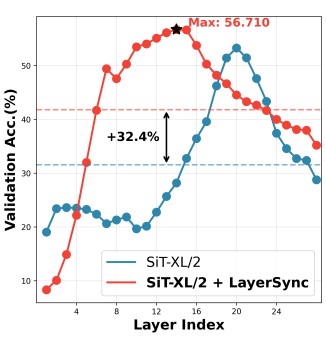

(a) Classification on Tiny ImageNet dataset.

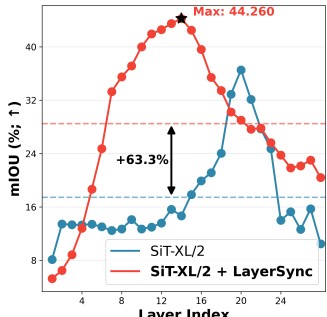

(b) Semantic Segmentation on PASCAL VOC dataset.

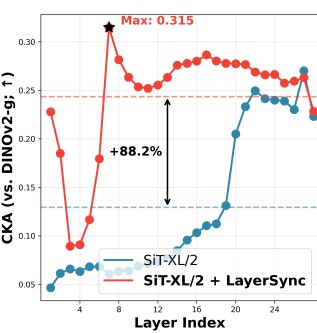

(c) Alignment to DINOv2-g.

Figure 4: Assessing the quality of intermediate features shows that LayerSync improves average validation accuracy across layers (shown with dashed lines in the figures) for both classification and segmentation, and enhances alignment with DINOv2. In this experiment, layer 8 is aligned with layer 16.

as a sequence of actions, the task is to generate the corresponding human motion. Each motion sequence consists of a series of human poses, where each pose is represented by 22 joints defined as 3D points in space.

**Implementation details.** We follow the exact setup as MDM (Tevet et al., 2022) using a transformer with 8 layers. We use HumanML3D dataset (Guo et al., 2022). We train the model with and without LayerSync for 600K iterations. We align layer 3 with 6. More details are provided in Section M

**Evaluation metrics.** We report FID and R-Precision (top 3) (Kynkäänniemi et al., 2019) as detailed in Section O.

**Results.** The results summarized in Table 4 show that LayerSync improves FID by 7.7% and R-Precision by 3.4%, confirming its effectiveness even with small architectures and limited datasets.

## 4.4 REPRESENTATION LEARNING

To evaluate the effect of LayerSync, we analyze the model's intermediate representations. We compare SiT-XL/2 model trained with LayerSync for **160 epochs** against a baseline SiT-XL/2 trained for **1400 epochs** as they both have similar FID. This ensures that both models exhibit comparable generative performance, allowing us to isolate the impact of our regularization on the learned representations, independently of the final generation quality.

We consider linear probing for classification on Tiny ImageNet dataset (Deng et al., 2009), linear probing for segmentation on the PASCAL VOC dataset (Everingham et al., 2010), and Centered Kernel Alignment (CKA) (Kornblith et al., 2019) with DINOv2 embeddings (Oquab et al., 2023) to measure the distance between the model representations. More implementation details are provided in Section L.

Our empirical results, summarized in Figure 4, lead to two interesting observations. First, LayerSync induces a more homogeneous distribution of high-quality features across the network's layers, leading to 32.4 % improvement in the average validation accuracy for classification, 63.3 % in average mIOU, and 88.2 % improvement in average alignment with DINOv2. Secondly, we observe not only a shift in the block with the best performance in downstream tasks but also an improvement in

Table 5: **Quantitative evaluation of SiT-XL/2 trained on ImageNet** $256 \times 256$ **(Deng et al., 2009) comparing REPA and REPA + LayerSync.** Both models are trained for 200k iterations and a batch size of 1024. REPA is applied on Layer 10 and LayerSync aligns layer 8 with 16. The results show that LayerSync can be combined with approaches that rely on external representations, such as REPA, to further accelerate training.

| Method | FID ↓ | sFID ↓ | Inception Score ↑ |
|---|---|---|---|
| REPA | 7.88 | 4.81 | 126.39 |
| REPA + LayerSync | **7.01** | **4.78** | **129.85** |

the best performing block. While an increase in mean performance is an intuitive consequence of regularizing weaker layers toward a high-performing one, the emergence of a new peak that significantly surpasses the baseline's maximum is a non-trivial finding.

We conclude that the representational benefits of LayerSync are not merely a byproduct of accelerated convergence. Even when the baseline model is afforded more than 8x larger training budget to match generative performance, its internal representations remain significantly inferior. We therefore hypothesize that LayerSync acts as a powerful structural regularizer that fundamentally alters the model's optimization trajectory. By imposing an internal semantic constraint, it guides the network to discover a more efficient and globally coherent feature hierarchy, one that remains inaccessible to the unconstrained model.

Furthermore, it is noteworthy that the quality of representations learned with LayerSync approaches that of models trained with powerful external guidance. Our peak classification accuracy, for example, is comparable to the results reported by (Yu et al., 2024). We interpret this as evidence for our initial hypothesis: LayerSync establishes a virtuous cycle that progressively refines the entire feature hierarchy. Notably, the layer of peak performance often shifts to align with the chosen reference layer. A more detailed analysis is provided in Section G.

## 4.5 LAYERSYNC COMBINED WITH EXTERNAL REPRESENTATIONS

In this experiment, we investigate whether LayerSync can be combined with external representation guidance to further accelerate training. The results summarized in Table 5 show that the two approaches are synergistic; combining LayerSync with REPA (Yu et al., 2024) yields better performance than REPA alone. This suggests that the internal structural alignment of LayerSync (see Section 3.2) and the external semantic injection of REPA operate as complementary axes of improvement. Thus, LayerSync can be used as a standalone method when external models are unavailable or too costly, or combined with external representations to maximize performance. Additional details provided in Section H

## 5 ABLATION STUDY

**Layer Selection.** To empirically validate the robustness of our layer selection strategy, we conducted an experiment with randomized layer pairings. For both SiT-XL and SiT-L architectures, detailed in Tables 11 and 12. We observe that regardless of the selected layer, we always have a gain. However, the gain is optimal if following the layer selection strategy discussed in Section 3.4. The results in Table 6 demonstrate remarkable consistency. The low standard deviation in the FID (0.8 for SiT-XL) confirms that the specific choice of layers is not a very sensitive hyperparameter. This robustness validates our claim that LayerSync is a practical, plug-and-play method that provides significant performance gains without necessitating an expensive search for optimal layer combinations.

**Effect of $\lambda$.** We examine the effect of the regularization coefficient $\lambda$ on SIT B/2 in Table 7 and observe that our method is robust to a wide range of values for $\lambda$ and consistently improves FID.

Table 6: Performance of LayerSync with randomized layer pairings. Results show the mean FID, IS and standard deviation (in parentheses) over independent runs, confirming the robustness of Layer-Sync to layer selection.

| Method | Model | FID ↓ (STD) | IS ↑ (STD) |
|---|---|---|---|
| Baseline | SiT-XL | 17.98 | 50.41 |
| **LayerSync - Ours** | SiT-XL | **12.24 (0.8)** | **71.47 (2.6)** |

Table 7: Ablation study for $\lambda$. We train SiT-B/2 for 400K iterations while aligning block 2 with 8. We observe that our approach is robust for a wide range of $\lambda$. The baseline SiT-B/2 has FID 36.19.

| $\lambda$ | 0 | 0.1 | 0.2 | 0.3 | 0.5 | 0.7 | Average (Std) |
|---|---|---|---|---|---|---|---|
| FID↓ | 36.19 | 31.63 | 31.02 | 31.6 | 31.17 | 31.36 | 31.356 (0.27) |
| IS↑ | - | 44.9 | 46.12 | 44.56 | 45.65 | 45.2 | 45.286 (0.61) |

## 6 DISCUSSION

LayerSync is a regularization framework that promotes feature consistency across a model's depth. It aligns intermediate layers by encouraging those with weaker representations to become more similar but not identical to those with richer features. This self-alignment propagates strong semantic information, which we found accelerates training and improves generative performance.

This similarity between layers raises a natural question: does LayerSync make layers redundant, potentially allowing for model pruning?

Our experiments (Appendix B) show that while models trained with LayerSync are more robust to layer removal than their baseline counterparts, performance still degrades significantly. This indicates that despite the improved alignment, each layer retains a unique function essential to the model's capacity. Consequently, naively pruning a trained model did not prove superior to simply training a smaller architecture from scratch. Similar findings have been reported for LLMs: intermediate transformer blocks often exhibit high correlation. While removing these blocks has little effect on easier tasks such as question answering (QA), it degrades performance on more challenging tasks (Gromov et al., 2024). Therefore, high inter-block correlation does not necessarily imply that the blocks are redundant.

We also wish to emphasize that the long-term effects of regularization might also demand further study. Although we did not observe the performance degradation seen in other methods with external guidance as reported in (Wang et al., 2025), future work could explore scheduling the LayerSync loss to preemptively address any potential long-term downsides.

Finally, the alignment loss function itself presents a key area for future research. We selected cosine similarity due to its strong empirical performance on images and its effective transfer to audio. However, developing novel alignment losses specifically engineered for different data domains, such as the hierarchical nature of text or the temporal patterns in time-series data, is an interesting and potentially impactful research direction.

## 7 CONCLUSION

In this paper, we introduced LayerSync, a simple yet novel self-supervised regularization method for improving diffusion transformers. We demonstrated that a model's later-layer representations can effectively guide its earlier layers, enhancing feature quality and accelerating training at no additional cost. As a general framework, LayerSync requires no external guidance and is readily applicable to different data domains.

This work opens several avenues for future research in training efficiency, representation learning, and self-supervised learning. We believe the core principle of LayerSync is broadly applicable and encourage exploring its potential in other generative architectures beyond diffusion models.

## 8 REPRODUCIBILITY STATEMENT

We are committed to open-sourcing the full codebase and all experiment configurations used in this paper upon acceptance, to support transparency and facilitate future research in this area.

## 9 ETHICS STATEMENT

This work investigates a self-supervised method for accelerating the training of diffusion models across multiple modalities. All experiments were conducted using publicly available datasets. No personal or sensitive data was used, and we do not anticipate any direct ethical risks associated with this research.

## 10 ACKNOWLEDGMENT

This research was supported by the Swiss National Science Foundation (SNSF) under Grant No. 10003100, and by Innosuisse – the Swiss Innovation Agency (Ref. 127.265 IP-ICT). Computational resources were provided as a part of the Swiss AI Initiative by a grant from the Swiss National Supercomputing Centre (CSCS) under project ID a144 on Alps.

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

# A  VIDEO GENERATION

We study the effectiveness of LayerSync for both video generation and fine-tuning existing video diffusion models on new datasets.

**Training from Scratch on CLEVRER.** We train SiT-XL model with 3D patchification of size (1,2,2) on the CLEVRER dataset (Yi et al., 2019). Due to the high computational cost of video training from scratch, we limit the run to 24k steps on 16 GPUs, using this as a proof of concept to demonstrate the effectiveness of LayerSync.

**Fine-tuning on MixKit dataset.** We use MixKit dataset (Lin et al., 2024) to fine-tune Wan2.1 1.3B foundation model (Wan et al., 2025). Each video has 81 frames. We align layer 8 with layer 16 with a regularizer weight of 0.02. The model is fine-tuned via LoRA adapters for 1,100 steps using 4 GPUs.

**Baselines.** We refer to fine-tuning without any extra guidance as vanilla.

**Evaluation metrics.** We rely on Fréchet Video Distance (FVD; Unterthiner et al. (2018)) for evaluation. We generate 5,000 videos of 16 frames for SiT-XL and 1,000 videos of 81 frames for the fine-tuned Wan2.1 model.

**Results.** As shown in Table 8, LayerSync consistently outperforms vanilla training across both fine-tuning and training from-scratch setups, achieving the lowest FVD scores in every scenario.

Table 8: **FVD scores (↓) for video generation.** We observe that LayerSync consistently improves FVD for both fine-tuning and training from scratch.

|  | Wan2.1 (MixKit) | SiT-XL (CLEVRER) |
|---|---|---|
| Vanilla | 321.84 | 265.50 |
| **LayerSync (Ours)** | **304.68** | **120.13** |

# B  DROPPING BLOCKS

To investigate the effect of dropping blocks, we train SiT XL/2 and SiT XL/2 with LayerSync (aligning layers 7 and 16) for 120k iterations. We then drop four blocks in between the aligned layers. Quantitative results are presented in Table 9, and qualitative examples are shown in Figure 5, indicating that the model trained with LayerSync is more robust to block removal.

We also experimented dropping blocks outside the aligned layers. As summarized in Table 9 and Figure 6, this leads to a more significant degradation in sample quality, suggesting that the drop of blocks outside the synced range has a more detrimental effect. Although LayerSync improves robustness to dropped blocks, doing so still results in an increase in FID.

Table 9: Comparison of FID, sFID, Inception Score (IS), Precision, and Recall when dropping specific blocks from the model. The model trained with LayerSync is more robust to block removal.

|  | Skipped blocks | FID ↓ | sFID ↓ | IS ↑ | Precision ↑ | Recall ↑ |
|---|---|---|---|---|---|---|
| SiT XL/2 | - | 37.03 | 5.49 | 35.41 | 0.53 | **0.61** |
| SiT XL/2 + LayerSync | - | **25.72** | **5.05** | **48.49** | **0.61** | 0.59 |
| SiT XL/2 | [9,11,13,15] | 211.66 | 93.92 | 4.02 | 0.01 | 0.10 |
| SiT XL/2 + LayerSync | [9,11,13,15] | 55.07 | 7.85 | 23.04 | 0.39 | 0.63 |
| SiT XL/2 + LayerSync | [9,11,13,15,21] | 86.11 | 18.30 | 16.79 | 0.29 | 0.46 |
| SiT XL/2 + LayerSync | [1,9,11,13,15] | 92.84 | 22.28 | 15.38 | 0.26 | 0.44 |

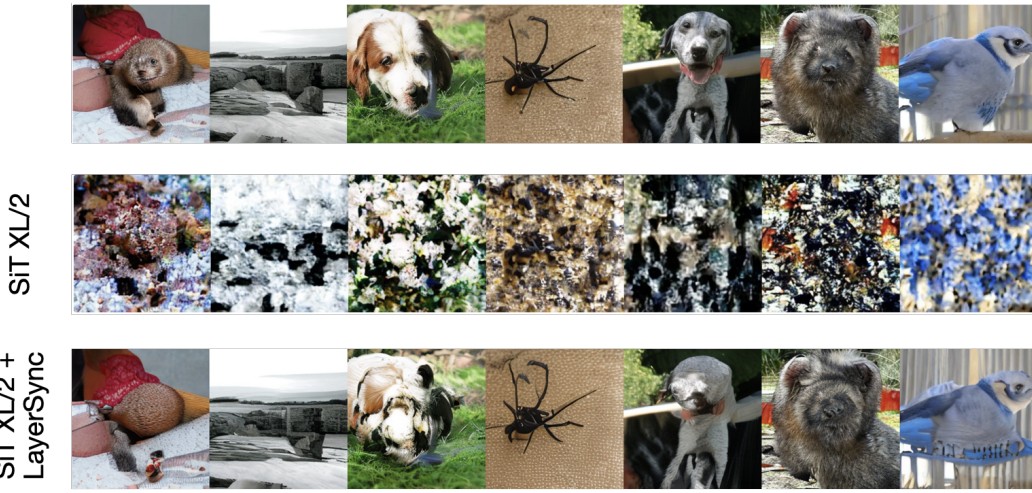

Figure 5: Qualitative comparison of generated samples from SiT XL/2 and SiT XL/2 with LayerSync when layers 7 and 16 are synced. After dropping blocks [9, 11, 13, 15], we observe that LayerSync helps preserve visual quality despite block removal.

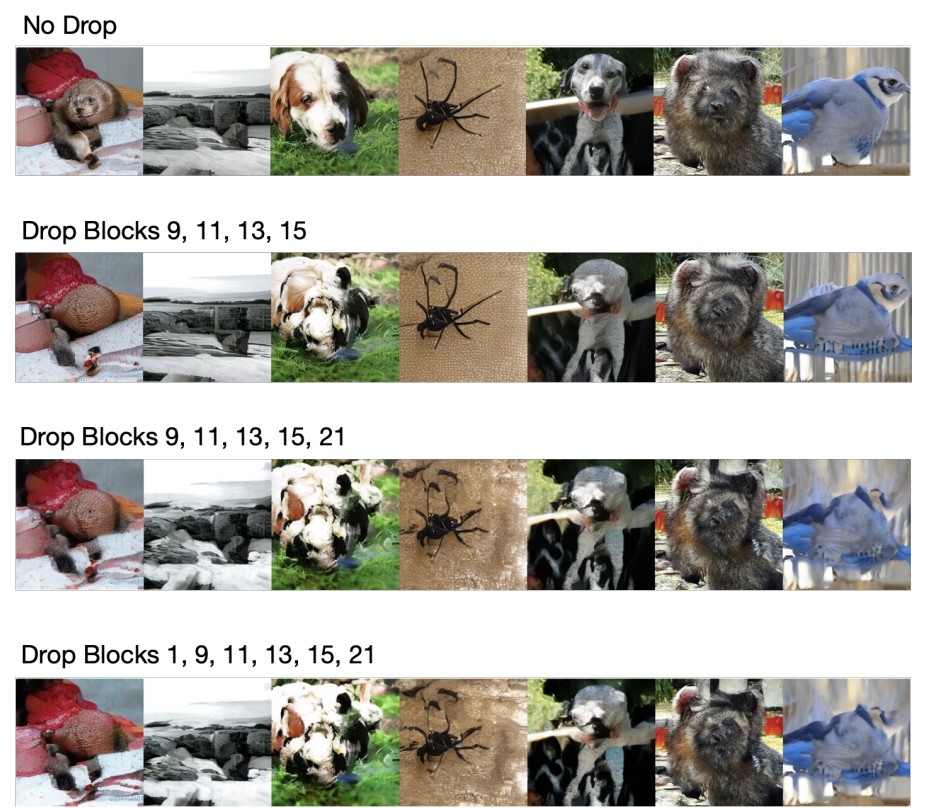

Figure 6: Qualitative results when dropping blocks from SiT XL/2 with LayerSync, where layers 7 and 16 are synced. Dropping blocks outside the synced layers leads to a more noticeable degradation in sample quality.

## C  ALIGNMENT CONSIDERING THE TIMESTEP

We apply alignment on SiTXL/2 for the last 75 %, 50 % and 25 % of timesteps. The model is trained on ImageNet $256 \times 256$ for 80 epochs. The results summarized in Table 10 shows that the best performance is achieved when alignment is applied on all the timesteps, which confirms that improving the weak representations is beneficial regardless of the timestep.

Table 10: Applying LayerSync on specific timesteps shows that alignment is beneficial for all the timesteps.

| Timestep | FID ↓ |
|----------|-------|
| 25% | 18.28 |
| 50% | 18.77 |
| 75% | 17.68 |
| 100% | **16.03** |

## D  ABLATION ON BLOCK SELECTION

Ablation study on block selection for SiT-XL/2 and SiT-L/2 summarized in Tables 11 and 12 shows that LayerSync consistently improves the generation quality, but the gain is suboptimal when the distance between the blocks is low or when aligning with the decoder blocks (very last layers). All the models are trained for 100K iterations with 16 GPUs and a batch size of 1024.

Table 11: **Ablation study on block selection for SiT-XL/2.** The results show that LayerSync consistently improves the generation quality, but the gain is suboptimal when the distance between the blocks is low or when aligning with the decoder blocks (very last layers).

| Model | IS ↑ | FID ↓ | sFID ↓ |
|-------|------|-------|--------|
| Vanilla SiT-XL/2 | 50.408 | 26.534 | 5.035 |
| LayerSync 6 ← 18 | 75.221 | 15.386 | 4.672 |
| LayerSync 7 ← 17 | 75.727 | 15.433 | 4.694 |
| LayerSync 7 ← 15 | 74.987 | 15.740 | 4.611 |
| LayerSync 9 ← 17 | 73.791 | 16.058 | 4.695 |
| LayerSync 8 ← 18 | 73.549 | 16.078 | 4.589 |
| LayerSync 8 ← 17 | 72.821 | 16.276 | 4.638 |
| LayerSync 8 ← 20 | 71.477 | 16.568 | 4.662 |
| LayerSync 9 ← 20 | 71.016 | 16.661 | 4.695 |
| LayerSync 8 ← 19 | 71.094 | 16.680 | 4.658 |
| LayerSync 6 ← 14 | 72.750 | 16.697 | 4.695 |
| LayerSync 10 ← 18 | 70.209 | 16.999 | 4.638 |
| LayerSync 9 ← 19 | 69.831 | 17.091 | 4.701 |
| LayerSync 11 ← 21 | 68.952 | 17.096 | 4.673 |
| LayerSync 7 ← 21 | 69.438 | 17.126 | 4.685 |
| LayerSync 11 ← 19 | 68.464 | 17.615 | 4.628 |
| LayerSync 9 ← 22 | 68.127 | 17.823 | 4.717 |
| LayerSync 8 ← 21 | 67.633 | 18.032 | 4.705 |
| LayerSync 6 ← 8 | 57.956 | 22.630 | 4.903 |
| LayerSync 6 ← 9 | 56.552 | 23.307 | 4.892 |
| LayerSync 6 ← 7 | 55.717 | 23.924 | 4.954 |
| LayerSync 11 ← 23 | 61.851 | 19.966 | 4.769 |
| LayerSync 12 ← 23 | 59.692 | 20.711 | 4.737 |
| LayerSync 15 ← 23 | 55.235 | 23.030 | 4.783 |

Table 12: **Ablation study on block selection for SiT-L/2.** The results show that LayerSync consistently improves the generation quality, but the gain is suboptimal when the distance between the blocks is low or when aligning with the decoder blocks (very last layers).

| Model | IS | FID | sFID |
|---|---|---|---|
| LayerSync 6 ← 15 | 64.638 | 19.165 | 4.827 |
| LayerSync 8 ← 16 | 62.794 | 19.566 | 4.856 |
| LayerSync 7 ← 15 | 62.854 | 19.663 | 4.841 |
| LayerSync 6 ← 14 | 62.616 | 19.945 | 4.791 |
| LayerSync 5 ← 13 | 62.158 | 20.048 | 4.778 |
| LayerSync 7 ← 16 | 61.4 | 20.055 | 4.865 |
| LayerSync 7 ← 18 | 60.533 | 20.324 | 4.820 |
| LayerSync 6 ← 16 | 61.304 | 20.444 | 4.794 |
| LayerSync 5 ← 12 | 60.807 | 20.492 | 4.775 |
| LayerSync 5 ← 11 | 61.104 | 20.696 | 4.821 |
| LayerSync 5 ← 15 | 60.362 | 20.734 | 4.824 |
| LayerSync 5 ← 17 | 61.178 | 20.771 | 4.878 |
| LayerSync 5 ← 8 | 50.473 | 26.511 | 5.084 |
| LayerSync 5 ← 7 | 49.948 | 27.004 | 5.171 |
| LayerSync 5 ← 6 | 47.19 | 28.741 | 5.21 |
| LayerSync 11 ← 20 | 43.253 | 30.57 | 5.172 |
| LayerSync 17 ← 20 | 43.563 | 31.056 | 5.308 |
| LayerSync 16 ← 20 | 43.242 | 31.24 | 5.316 |
| LayerSync 13 ← 20 | 42.535 | 31.407 | 5.182 |
| LayerSync 14 ← 20 | 41.626 | 32.35 | 5.282 |
| LayerSync 15 ← 20 | 41.381 | 32.622 | 5.403 |

# E    LAYERSYNC VS. INCREASING LEARNING RATE

To show that LayerSync impact is not simply due to an increase in the gradient magnitude, we designed two sets of experiments below. All the models are trained with 16 GPUs, batch size 1024 for 100k iterations (80 epochs). We then report the gradient norms and the FIDs of different configurations. We consistently observed that the gradient norm with LayerSync is actually smaller than that of the baseline, indicating that our method is not simply equivalent to using a larger learning rate.

**(1) Global learning rate increase.**

Starting from the default learning rate of $1 \times 10^{-4}$, we trained models with higher learning rates $2 \times 10^{-4}$ and $5 \times 10^{-4}$. For learning rates above $5 \times 10^{-4}$, the model diverges. While increasing the learning rate can partially accelerate training, the resulting FID scores remain worse than those obtained with LayerSync with the default learning rate $1 \times 10^{-4}$ as shown in Table 13. The visualization of gradient norm is provided in Figure 7a.

Table 13: **Effect of Global Learning Rate.** Simply increasing the global learning rate improves FID slightly but does not match the performance gains of LayerSync.

| Method | lr. | FID |
|---|---|---|
| SiT-XL/2 | $1 \times 10^{-4}$ | 26.53 |
| SiT-XL/2 | $2 \times 10^{-4}$ | 24.95 |
| LayerSync | $1 \times 10^{-4}$ | 16.03 |

**(2) Higher learning rate on early blocks only.**

We then increased the learning rate only for the first 8 blocks and compared this to a model trained with LayerSync aligning layers 8–16 at a global learning rate of $1 \times 10^{-4}$. Again, for learning rates above $1 \times 10^{-3}$ training diverges. As summarized in Table 14, for $2 \times 10^{-4}$, $5 \times 10^{-4}$, and $1 \times 10^{-3}$, the FID improvements do not match those obtained with LayerSync. The visualization of the gradient norms is provided in Figure 7b.

Table 14: **Effect of Early-Layer Learning Rate.** Increasing the learning rate specifically on early layers (first 8 blocks) is beneficial but still underperforms compared to LayerSync.

| Method | lr. Early Layers | General lr. | FID |
|---|---|---|---|
| SiT-XL/2 | $1 \times 10^{-4}$ | $1 \times 10^{-4}$ | 26.53 |
| SiT-XL/2 | $2 \times 10^{-4}$ | $1 \times 10^{-4}$ | 19.24 |
| SiT-XL/2 | $5 \times 10^{-4}$ | $1 \times 10^{-4}$ | 24.63 |
| LayerSync | $1 \times 10^{-4}$ | $1 \times 10^{-4}$ | 16.03 |

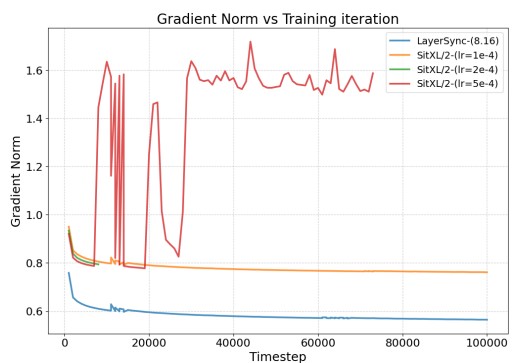

(a) Gradient norm for different learning rates.

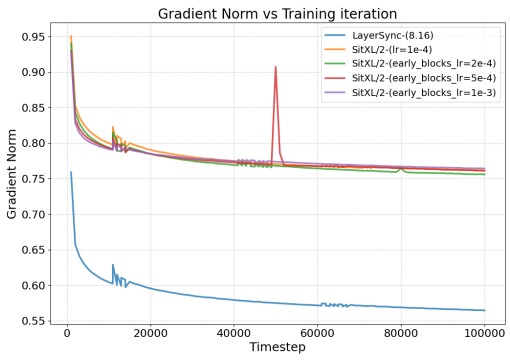

(b) Gradient norm when changing the learning rate of early blocks.

Figure 7: **Gradient norm visualization revealed that LayerSync impact is not similar to simply increasing the learning rate as the model trained with LayerSync has lower gradient norm than the baseline.**

## F  LAYERSYNC VS SELF-REPRESENTATION ALIGNMENT USING EMA

We compare our LayerSync approach with the concurrent work SRA (Jiang et al., 2025) in terms of training time and computational overhead. We computed the metrics for SiT-XL/2 using 4 GH200 GPUs and a batch size of 32 per GPU. Results are reported in Table 15. We show that LayerSync requires $25.5\%$ fewer Flops, is $40.5\%$ faster in real-time, and reaches an FID $5\%$ higher.

Table 15: **Comparison between LayerSync and SRA.** LayerSync results in lower FID while being less computationally expensive.

| Method | FID $\downarrow$ | Wall-clock time/step $\downarrow$ | GFlops $\downarrow$ |
|---|---|---|---|
| SiT-XL/2+SRA | 1.58 | 0.617 | 30762 |
| SiT-XL/2+LayerSync | **1.49** | **0.367** | **22910** |

## G  THE VIRTUOUS CYCLE AND THE EVOLUTION OF REPRESENTATIONS

To provide empirical support for the "virtuous cycle" and analyze the evolution of the feature hierarchy, we conducted two additional studies evaluating the segmentation performance (mIoU) of internal representations throughout the network on PASCAL VOC (Everingham et al., 2010).

First, we tracked the evolution of representations throughout the training process (from 100k to 600k steps). As shown in Figure 8a, we observe that while the relative structure of feature quality across layers remains stable, the performance monotonically improves across the entire hierarchy.

Our second study evaluates models trained with different alignment targets at the same training stage (100k steps) (see Figure 8b). This comparison provides the most compelling evidence for our hypothesis. When we synchronize an early block (e.g., block 6) with a deeper target (block 18 vs. block 10), we observe two critical effects:

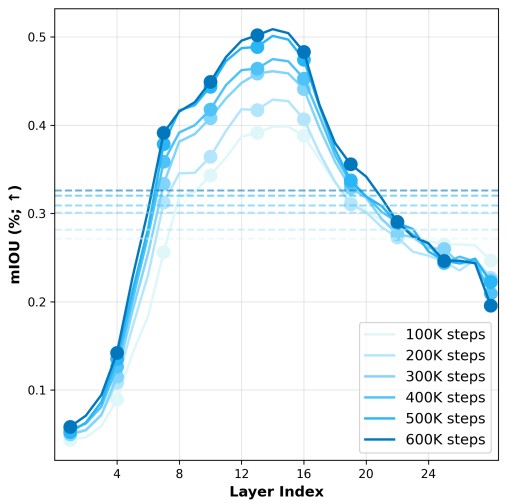 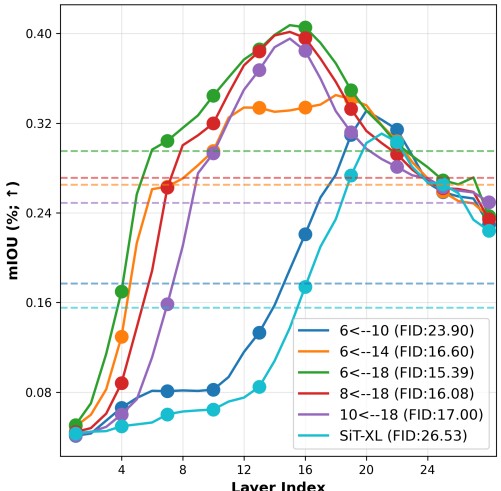

(a) Evolution of representations throughout the training process assessed by semantic segmentation on PASCAL VOC dataset.

(b) Evaluation of representations for different layers alignments assessed by semantic segmentation on PASCAL VOC dataset.

Figure 8: **Evaluation of the representations assessed by semantic segmentation on PASCAL VOC dataset.** (a) Evolution throughout the training process. (b) Evaluation of representation for different layer alignment.

- **Global Improvement:** The model guided by the deeper layer achieves superior downstream performance and lower FID (15.39 vs. 23.90), indicating that guidance from deeper layers correlates with better overall representations.

- **Accelerated Maturation:** Notably, using a deeper target shifts the peak performance of the network to earlier layers. By effectively "pulling" semantic richness from the deep target to the earlier block, the early layers appear to acquire higher-level features sooner in the depth hierarchy.

These observations are consistent with the hypothesized virtuous cycle: guiding an early block with a stronger target improves its representation, which in turn provides higher-quality input to subsequent layers. This likely facilitates the learning of stronger deep representations, which then serve as even better guides, progressively refining the entire hierarchy.

## H  OPTIMAL PLACEMENT OF EXTERNAL GUIDANCE WHEN COMBINED WITH LAYERSYNC

To maximize the synergy between internal and external alignment, we investigated the optimal depth for applying REPA. The results are summarized in Table 16 and show that applying REPA *before* the synchronization range leads to no significant synergies. The most effective strategy integrates the external signal *between* the aligned layers. For instance, using LayerSync to align layers 8 and 16 while applying REPA at layer 10 yields the best performance. All the models are trained for 50k iterations with 16 GPUs and a batch size of 1024. To show that the trend continues, we continue training the models to 200k iterations and report the results in the main paper.

## I  EVOLUTION OF BLOCK STRUCTURE

A comparison between the block structure of the SiT-XL/2 and SiT-XL/2 + LayerSync is provided in Figure 9, showing that LayerSync imposes the structural equilibrium early in training.

Table 16: **Qualitative comparison between different combinations of REPA and LayerSync.** The results show that combining LayerSync with REPA can further accelerate the training, and the best place to apply REPA is between the syncing layers.

| Method | REPA Layer | LayerSync Layer | FID $\downarrow$ |
|---|---|---|---|
| SiTXL/2 | – | – | 59.45 |
| SiTXL/2 + LayerSync | – | 8–16 | 46.26 |
| SiTXL/2 + REPA | 7 | – | 46.06 |
| SiTXL/2 + REPA + LayerSync | 7 | 8–16 | 43.55 |
| SiTXL/2 + REPA + LayerSync | 10 | 8–16 | **29.68** |

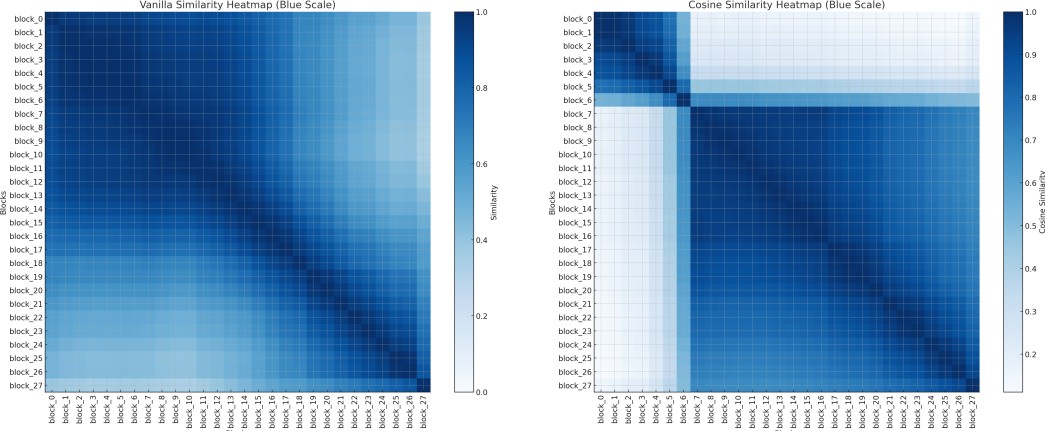

(a) Correlation between the blocks of SiT-XL/2 after 120k iterations.

(b) Correlation between the blocks of SiT-XL/2 with LayerSync after 120k iterations.

Figure 9: **Evolution of the Block Structure.** (a) Early in training (120k iterations), the vanilla model lacks this structure. (b) LayerSync imposes such structural equilibrium early in training (120k iterations).

## J   LAYERSYNC - ALGORITHM

We present the algorithmic formulation of LayerSync in Algorithm 1.

---
**Algorithm 1** LayerSync

---
**Require:** Weak Representation $Z_k \in \mathbb{R}^{B \times P \times D}$, Strong Representation $Z_{k'} \in \mathbb{R}^{B \times P \times D}$
      where $B$ is the batch size, $P$ the number of patches and $D$ the feature dimension.
 1: $Z_k^{\text{norm}} \leftarrow \text{normalize}(Z_k, \text{dim} = -1)$          $\triangleright$ L2-normalize embeddings
 2: $Z_{k'}^{\text{norm}} \leftarrow \text{normalize}(Z_{k'}, \text{dim} = -1)$          $\triangleright$ L2-normalize embeddings
 3: $\mathcal{L}_{\text{LayerSync}} \leftarrow -\text{similarity}(\sum_{j=1}^{P} Z_k^{\text{norm}}[:,j] \cdot Z_{k'}^{\text{norm}}[:,j])$    $\triangleright$ Negative similarity across patches
 4: **return** $\mathcal{L}_{\text{LayerSync}}$

---

## K   FLOP COMPARISON

We compare the computational complexity of Dispersive Loss, which computes pairwise distances, with LayerSync in Table 17. LayerSync is more efficient in terms of computational complexity, as the pairwise comparisons in Dispersive Loss result in a quadratic cost with respect to batch size.

Table 17: Comparison of computational complexity between the Dispersive Loss (pairwise distances) and LayerSync. $B$ is the batch size and $D$ is the feature dimension.

|  | FLOPs | Scaling w.r.t. Batch Size |
|---|---|---|
| Dispersive | $\mathcal{O}(B^2 D)$ | Quadratic ($B^2$) |
| LayerSync | $\mathcal{O}(BD)$ | Linear ($B$) |

## L    IMAGE GENERATION EXPERIMENTAL DETAILS

We use a node of 4 GH200 GPUs and a batch size of 256. The details of hyperparameters and sampler are provided in Tables 18 and 19.

**Classification.**    We use the Tiny ImageNet dataset (Deng et al., 2009), upsample the images to $256 \times 256$, and train linear classification heads for 50 epochs. Performance is evaluated on the validation set.

**Segmentation.**    For segmentation, we use the PASCAL VOC dataset (Everingham et al., 2010) and train linear heads for 25 epochs.

**CKA.**    For CKA evaluations (Kornblith et al., 2019), we use 4,000 samples from ImageNet $256 \times 256$.

Table 18: Hyperparameter setup for main experiments.

|  | Table 1 (SiT-B) | Table 1 (SiT-L) | Table 1 (SiT-XL) | Table 2 |
|---|---|---|---|---|
| **Architecture** |  |  |  |  |
| Input dim. | $32 \times 32 \times 4$ | $32 \times 32 \times 4$ | $32 \times 32 \times 4$ | $32 \times 32 \times 4$ |
| Num. layers | 12 | 24 | 28 | 28 |
| Hidden dim. | 768 | 1024 | 1152 | 1152 |
| Num. heads | 12 | 16 | 16 | 16 |
| **LayerSync** |  |  |  |  |
| $\lambda$ | 0.3 | 0.2 | 0.2 | 0.2 |
| Syncing layers | (4,7) | (8,18) | (8,16) | (8,16) |
| $\text{sim}(\cdot, \cdot)$ | cos. sim. | cos. sim. | cos. sim. | cos. sim. |
| **Optimization** |  |  |  |  |
| Batch size | 256 | 256 | 256 | 256 |
| Optimizer | AdamW | AdamW | AdamW | AdamW |
| lr | 0.0001 | 0.0001 | 0.0001 | 0.0001 |
| **Interpolants** |  |  |  |  |
| $\alpha_t$ | $t$ | $t$ | $t$ | $t$ |
| $\sigma_t$ | $1 - t$ | $1 - t$ | $1 - t$ | $1 - t$ |
| Training objective | v-prediction | v-prediction | v-prediction | v-prediction |
| Sampler | ODE Heun | ODE Heun | ODE Heun | SDE Euler–Maruyama |
| Sampling steps | 250 | 250 | 250 | 250 |
| Guidance | – | – | – | 1.37 |

## M    HUMAN MOTION GENERATION EXPERIMENTAL DETAILS

**Task.**    Given a sentence that describes a motion as a sequence of actions, the task is to generate a corresponding human motion. Each motion sequence consists of a series of human poses, where each pose is represented by 22 joints defined as 3D points in space.

Table 19: Hyperparameter setup for figures and ablation experiments.

| | Figure 3 (SiT-XL) | Table 6 (SiT-XL and SiT-B) | Table 7 (SiT-B) |
|---|---|---|---|
| **Architecture** | | | |
| Input dim. | $32 \times 32 \times 4$ | $32 \times 32 \times 4$ | $32 \times 32 \times 4$ |
| Num. layers | 28 | | 12 |
| Hidden dim. | 1152 | | 768 |
| Num. heads | 16 | | 12 |
| **LayerSync** | | | |
| $\lambda$ | 0.2 | 0.3 | - |
| Alignment depth | (8,16) | - | (2,8) |
| $\text{sim}(\cdot, \cdot)$ | cos. sim. | cos. sim. | cos. sim. |
| **Optimization** | | | |
| Training iteration | 400K | 400K | 400K |
| Batch size | 256 | 256 | 256 |
| Optimizer | AdamW | AdamW | AdamW |
| lr | 0.0001 | 0.0001 | 0.0001 |
| **Interpolants** | | | |
| $\alpha_t$ | $t$ | $t$ | $t$ |
| $\sigma_t$ | $1 - t$ | $1 - t$ | $1 - t$ |
| Training objective | v-prediction | v-prediction | v-prediction |
| Sampler | ODE Heun | ODE Heun | ODE Heun |
| Sampling steps | 250 | 250 | 250 |
| Guidance | – | – | – |

**Dataset.** We rely on HumanML3D dataset (Guo et al., 2022) that contains 44,970 motion annotations across 14,646 motion sequences from the AMASS (Mahmood et al., 2019) and HumanAct12 (Guo et al., 2020) datasets, along with corresponding text descriptions, and is widely used for the task of text-conditional human motion generation. Motions in the HumanML3D dataset follow the skeleton structure of SMPL (Loper et al., 2015) with 22 joints. Each pose $\mathbf{p}$ in the motion sequence is represented by a vector of size 237,

$$(r^a, \dot{r}^x, \dot{r}^z, r^y, j^p, j^v, j^r, c^f),$$

where $r^a \in \mathbb{R}$ is the root (pelvis joint) angular velocity along the Y-axis; $(\dot{r}^x, \dot{r}^z) \in \mathbb{R}$ are the root linear velocities in the XZ-plane; $r^y \in \mathbb{R}$ is the root height; $j^p \in \mathbb{R}^{3j}$, $j^v \in \mathbb{R}^{3j}$, and $j^r \in \mathbb{R}^{6j}$ are the local joint positions, velocities, and rotations in the root space, with $j$ indicating the number of joints; $c^f \in \mathbb{R}^4$ represents foot-ground contact features.

**Implementation details.** We use the exact setup as MDM (Tevet et al., 2022), we train up to 600K iterations using a H100 GPU. We sync block 3 with block 6.

## N   STOCHASTIC INTERPOLANTS

We adopt the generalized perspective of stochastic interpolants (Ma et al., 2024) which provides a unifying framework for both flow-based and diffusion-based models.

At the core of these models is a process that gradually transforms a real data sample $\mathbf{x}_0 \sim p(\mathbf{x})$ into a simple noise sample $\epsilon \sim \mathcal{N}(0, I)$. This process is defined by:

$$\mathbf{x}_t = \alpha_t \mathbf{x}_0 + \sigma_t \epsilon, \tag{4}$$

where $\alpha_t$ and $\sigma_t$ are functions of time, respectively decreasing and increasing, that control the mix of data and noise, satisfying the boundary conditions $\alpha_0 = \sigma_T = 1$, and $\alpha_T = \sigma_0 = 0$. The generative process aims to reverse this path. This can be model through a deterministic trajectory commonly described as the probability flow ordinary differential equation (PF-ODE).

$$\dot{\mathbf{x}}_t = v(\mathbf{x}_t, t), \tag{5}$$

where $v(\mathbf{x}_t, t)$ is the velocity field, specifying the direction and magnitude of movement at any point $\mathbf{x}_t$ at any time t to go from noise back to data. The velocity fields is defined as the time derivative of the interpolant:

$$v(\mathbf{x}, t) = \dot{\mathbf{x}}_t \big|_{\mathbf{x}_t = \mathbf{x}} = \dot{\alpha}_t \mathbb{E}[\mathbf{x}_0 \mid \mathbf{x}_t = \mathbf{x}] + \dot{\sigma}_t \mathbb{E}[\epsilon \mid \mathbf{x}_t = \mathbf{x}]. \tag{6}$$

However, since those conditional expectations are intractable, a model $v_\theta(\mathbf{x}_t, t)$ is trained to approximate it by minimizing the flow matching loss defined as:

$$\mathcal{L}_{\text{velocity}}(\theta) := \mathbb{E}_{\mathbf{x}_0, \epsilon, t} \left[ \|v_\theta(\mathbf{x}_t, t) - \dot{\alpha}_t \mathbf{x}_0 - \dot{\sigma}_t \epsilon\|^2 \right]. \tag{7}$$

The data is then generated by integrating equation 5 from t=1 to t=0 using any standard ODE solver starting from a random noise sample $\mathbf{x}_1 \sim \mathcal{N}(0, \mathbf{I})$. There exists also an alternative way to model the reverse process using Stochastic Differential Equation (SDE). The SDE shares the same marginal probability densities $p_t(\mathbf{x})$ as the PF-ODE but follows a stochastic, rather than deterministic, trajectory. The general form of this reverse SDE is:

$$d\mathbf{x}_t = \left( v(\mathbf{x}_t, t) - \frac{1}{2} w_t s(\mathbf{x}_t, t) \right) dt + \sqrt{w_t} d\mathbf{w}_t \tag{8}$$

where $w_t$ is a diffusion coefficient and $d\mathbf{w}_t$ is a standard Wiener process, and $s(\mathbf{x}_t, t)$ is the score function, defined as the gradient of the log-density of the data. The velocity and the score are not independent, they are two sides of the same coin as the score can be derived from the velocity field and vice versa.

## O  EVALUATION METRICS DETAILS.

### O.1  IMAGE

- **FID.** Heusel et al. (2017) measures the distance between the real and generated data distributions in the feature space of a pretrained Inception-v3 network (Szegedy et al., 2016). It computes the Fréchet distance (Heusel et al., 2017) between two multivariate Gaussians fitted to the feature embeddings, capturing both the quality and diversity of generated samples. Lower values indicate better performance.

- **sFID.** Nash et al. (2021) compares local image patches instead of global image statistics. By focusing on patch-level embeddings, sFID provides a more fine-grained evaluation of spatial consistency and local realism in the generated samples.

- **Inception Score.** Salimans et al. (2016) computes the Kullback–Leibler (KL) divergence (Kullback & Leibler, 1951) between conditional and marginal label distributions predicted by an Inception network.

- **Precision and Recall.** Kynkäänniemi et al. (2019) measures the fraction of generated samples that lie within the support of the real data distribution in feature space. High recall reflects the diversity of generated samples, indicating that the model captures the variability of the real data distribution.

### O.2  AUDIO

- **FAD**: Kilgour et al. (2018) like FID for images, is a reference-based metric that measures the perceptual similarity between the distribution of generated samples and the distribution of real audio.

### O.3 VIDEO

- **FVD**: Unterthiner et al. (2018) extends the idea of FID to videos by measuring the distance between real and generated video distributions in a pretrained spatiotemporal feature space. Specifically, it uses embeddings from Carreira & Zisserman (2017), pretrained on large-scale video datasets, to capture both spatial and temporal dynamics.

### O.4 MOTION

- **FID**: Computed in the same way as for images, but using T2M (Guo et al., 2022) motion features instead of Inception features.
- **R-Precision**: Measures the relevancy of the generated motions to the input prompts.

## P  EXTENDED RELATED WORK

In what follows, we summarize the main baseline methods used in our evaluation:

- **ADM** (Dhariwal & Nichol, 2021): Builds upon U-Net-based diffusion models by introducing classifier-guided sampling, allowing fine-grained control over the trade-off between generation quality and diversity.
- **VDM++** (Kingma & Gao, 2023): Proposes an adaptive noise schedule that adjusts dynamically during training, improving convergence and sample quality.
- **Simple diffusion** (Hoogeboom et al., 2023): Simplifies both the noise schedule and architectural components, enabling high-resolution image generation with improved computational efficiency.
- **CDM** (Ho et al., 2022): Introduces cascaded diffusion models that progressively refine images from low to high resolution using super-resolution stages, achieving better detail synthesis.
- **LDM** (Rombach et al., 2022): Trains diffusion models in a compressed latent space learned by a VAE, drastically reducing training cost while maintaining image fidelity.
- **U-ViT** (Bao et al., 2023): Combines ViT-based backbones with U-Net-style skip connections in the latent space, bridging the benefits of transformers and convolutional inductive biases.
- **DiffiT** (Hatamizadeh et al., 2024): Enhances transformer-based diffusion models using time-aware multi-head self-attention, boosting sample efficiency and reducing training time.
- **MDTv2** (Gao et al., 2023): Employs an asymmetric encoder-decoder transformer architecture with U-Net-inspired shortcuts in the encoder and dense skip connections in the decoder, improving video generation quality and coherence.
- **MaskDiT** (Zheng et al., 2023): Introduces masked modeling into diffusion transformers by training with an auxiliary mask reconstruction objective, leading to better efficiency and generalization.
- **SD-DiT** (Zhu et al., 2024): Builds on MaskDiT by incorporating a self-supervised discrimination objective using momentum encoding, enhancing the semantic richness of internal representations.
- **DiT** (Peebles & Xie, 2023): Proposes a pure transformer architecture for diffusion, using AdaLN-zero modules to stabilize training and scale to large model sizes efficiently.
- **SiT** (Ma et al., 2024): Investigates the link between training efficiency and flow-based perspectives by transitioning from discrete-time diffusion to continuous flow matching, showing improved sample quality and convergence rates.
- **D-JEPA** (Chen et al., 2024): Integrates Joint-Embedding Predictive Architectures (JEPA) into generative modeling by reframing masked image modeling as a generalized next-token prediction task, utilizing diffusion or flow matching loss to model per-token probability distributions in a continuous space.

- **VAR** (Tian et al., 2024): Redefines autoregressive image generation as a coarse-to-fine "next-scale prediction" process, diverging from standard raster-scan next-token prediction, allowing for faster inference and scaling laws similar to Large Language Models.
- **SRA** (Jiang et al., 2025): An EMA-based method that enables diffusion transformers to enhance their own representation learning and generation quality by aligning latent outputs from earlier, noisier layers with those from later, cleaner layers, eliminating the need for external guidance models.

## Q    DETAILS OF SiT MODEL

The architecture of the SiT block is provided in Figure 10, and more details on the model parameters are summarized in Table 20.

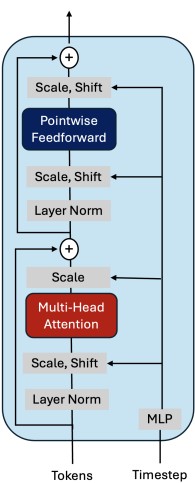

Figure 10: Visualization of a single SiT block.

Table 20: The number of transformer layers, hidden dimensionality, and number of attention heads for SiT models used in our experiments.

| Config | #Layers | Hidden dim | #Heads |
|--------|---------|------------|--------|
| B/2    | 12      | 768        | 12     |
| L/2    | 24      | 1024       | 16     |
| XL/2   | 28      | 1152       | 16     |

## R    ATTENTION MAPS PCA OVER LAYERS

We visualize the learned representations by applying PCA to the features of SiT-XL/2 models trained on ImageNet $256\times256$. We add different levels of noise to the input image and visualize the resulting features. We compare two variants: the baseline SiT-XL/2 and SiT-XL/2 with **LayerSync**, where block 8 is synced with block 16. Both models are trained for 400K iterations on a single node with 4 GH100 GPUs. Our results show that LayerSync results in more discriminative features, particularly in the earlier blocks.

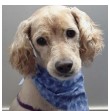

Figure 11: **Input image to the model**

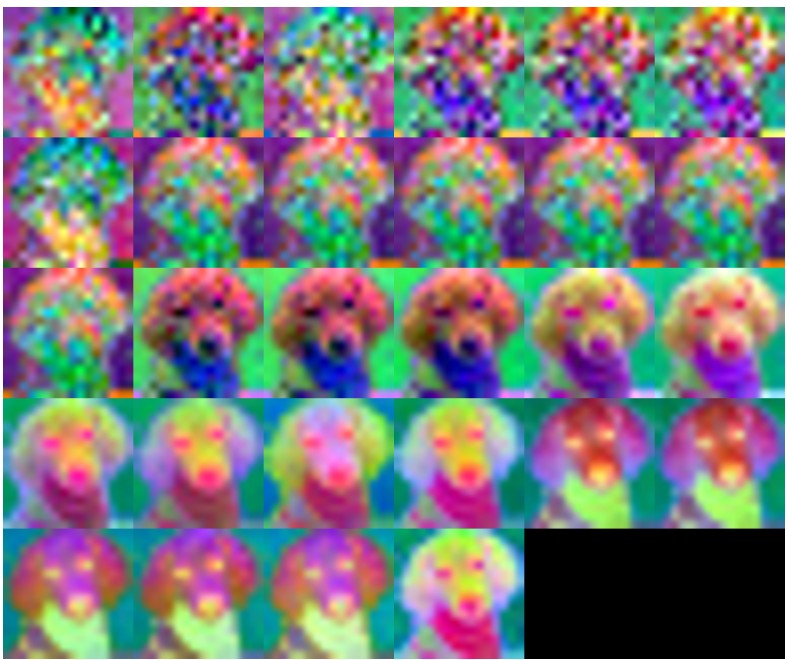

Figure 12: **Visualization of SiT-XL/2 model features with 10% noise added to the input image.** The top-left plot shows the features from the first block, and subsequent blocks are visualized row by row, ending with the final block in the bottom-right corner.

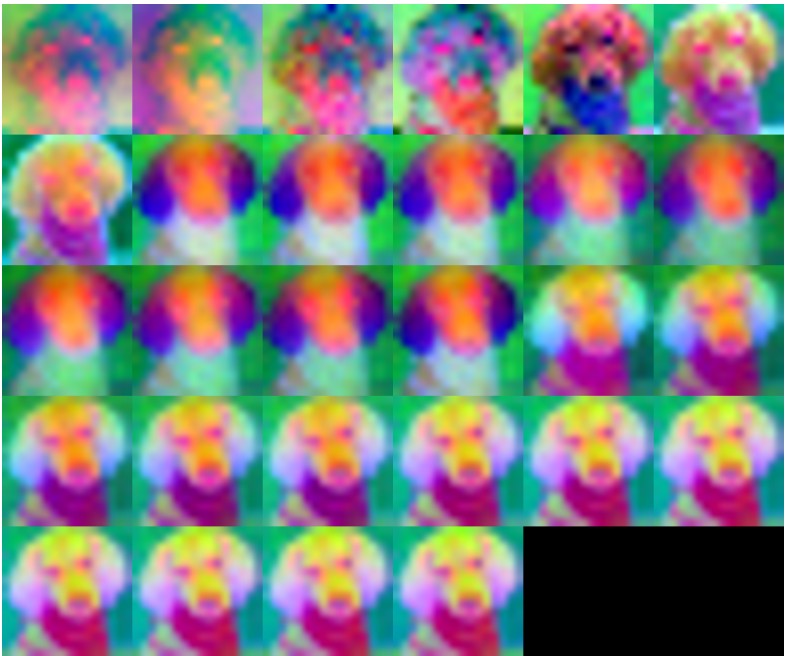

Figure 13: **Visualization of SiT-XL/2 model + LayerSync features with 10% noise added to the input image.** The top-left plot shows the features from the first block, and subsequent blocks are visualized row by row, ending with the final block in the bottom-right corner.

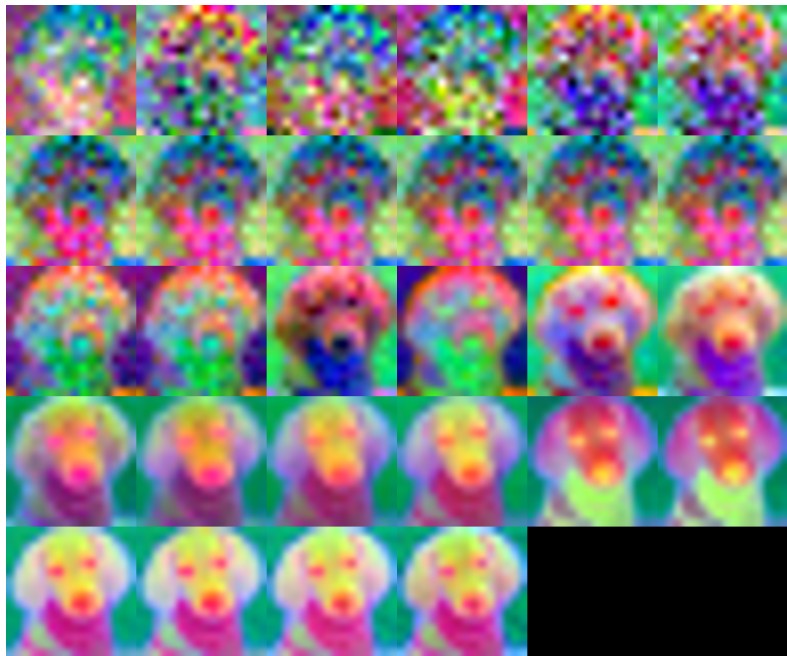

Figure 14: **Visualization of SiT-XL/2 model features with 30% noise added to the input image.** The top-left plot shows the features from the first block, and subsequent blocks are visualized row by row, ending with the final block in the bottom-right corner.

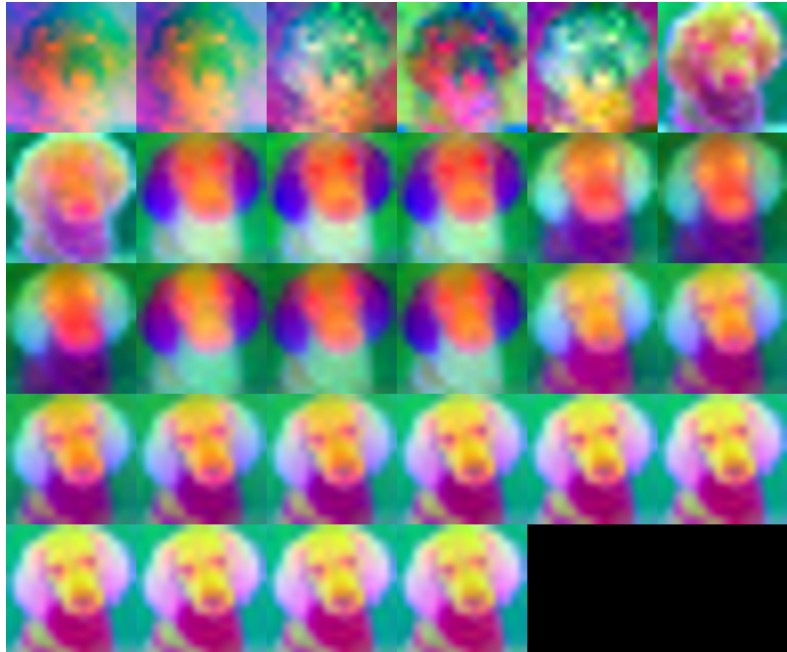

Figure 15: **Visualization of SiT-XL/2 model + LayerSync features with 30% noise added to the input image.** The top-left plot shows the features from the first block, and subsequent blocks are visualized row by row, ending with the final block in the bottom-right corner.

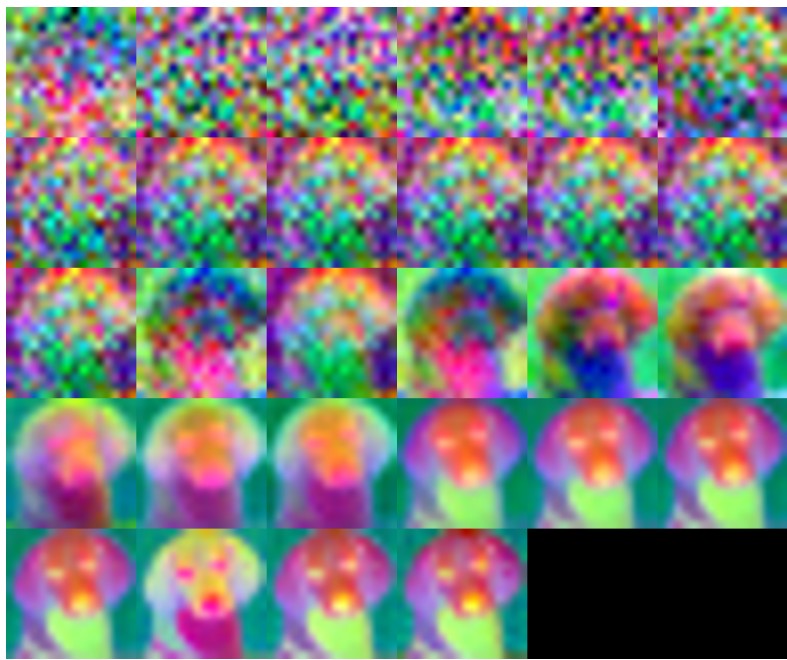

Figure 16: **Visualization of SiT-XL/2 model features with 50% noise added to the input image.** The top-left plot shows the features from the first block, and subsequent blocks are visualized row by row, ending with the final block in the bottom-right corner.

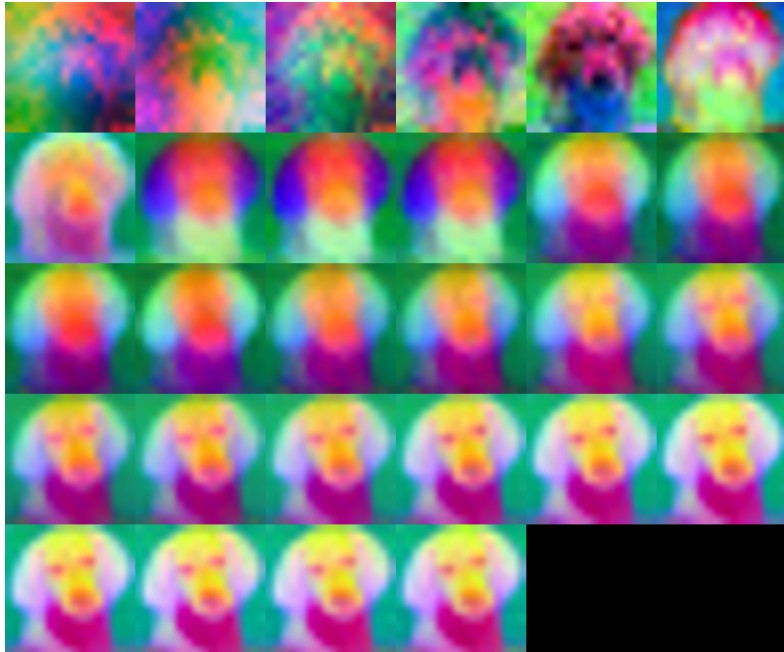

Figure 17: **Visualization of SiT-XL/2 model + LayerSync features with 50% noise added to the input image.** The top-left plot shows the features from the first block, and subsequent blocks are visualized row by row, ending with the final block in the bottom-right corner.

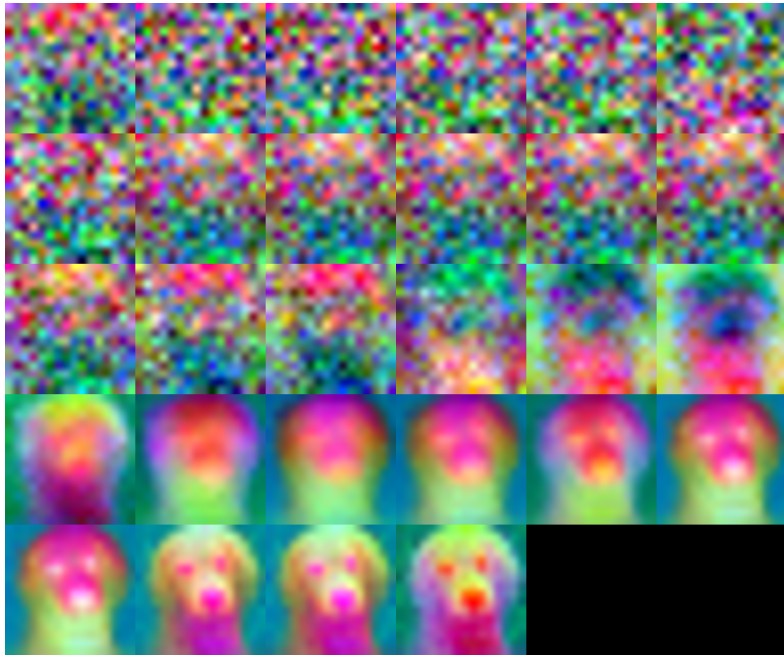

Figure 18: **Visualization of SiT-XL/2 model features with 70% noise added to the input image.** The top-left plot shows the features from the first block, and subsequent blocks are visualized row by row, ending with the final block in the bottom-right corner.

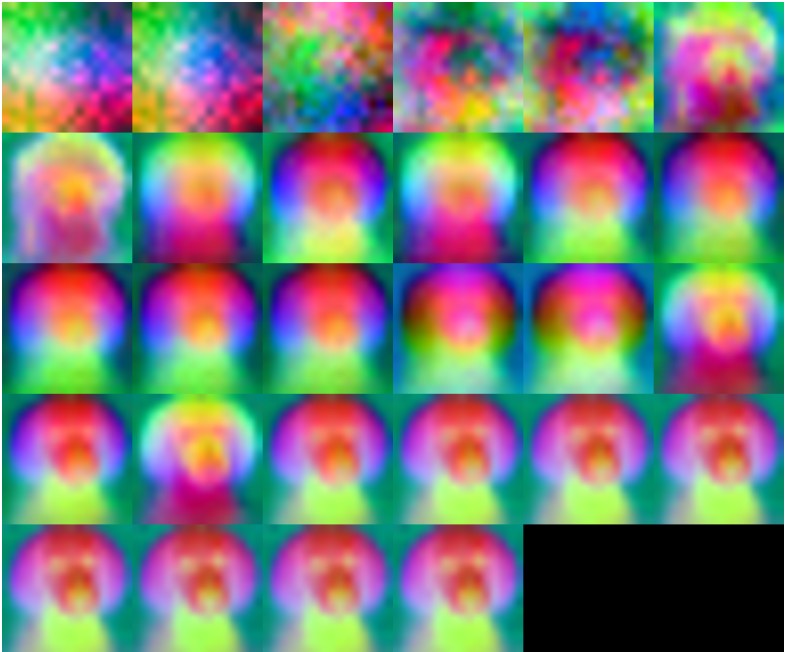

Figure 19: **Visualization of SiT-XL/2 model + LayerSync features with 70% noise added to the input image.** The top-left plot shows the features from the first block, and subsequent blocks are visualized row by row, ending with the final block in the bottom-right corner.

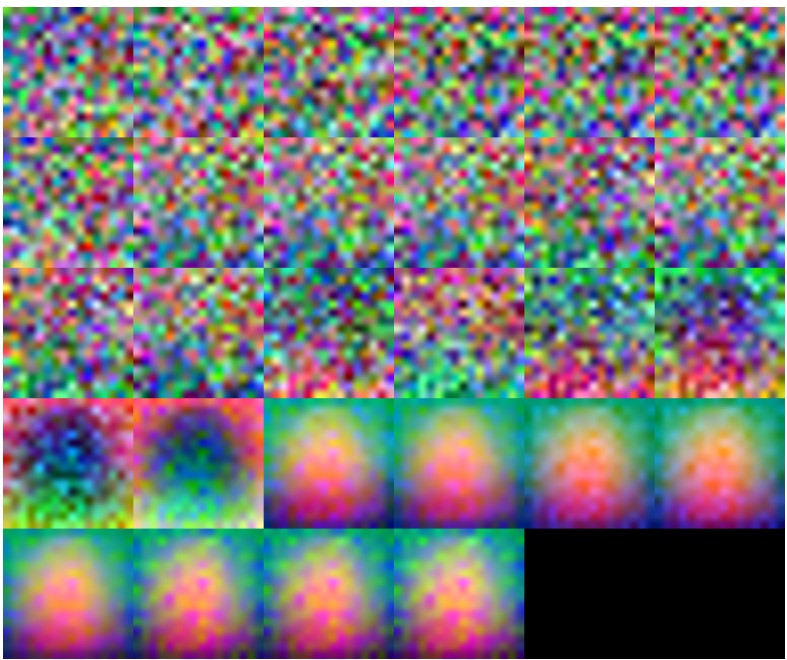

Figure 20: **Visualization of SiT-XL/2 model features with 90% noise added to the input image.** The top-left plot shows the features from the first block, and subsequent blocks are visualized row by row, ending with the final block in the bottom-right corner.

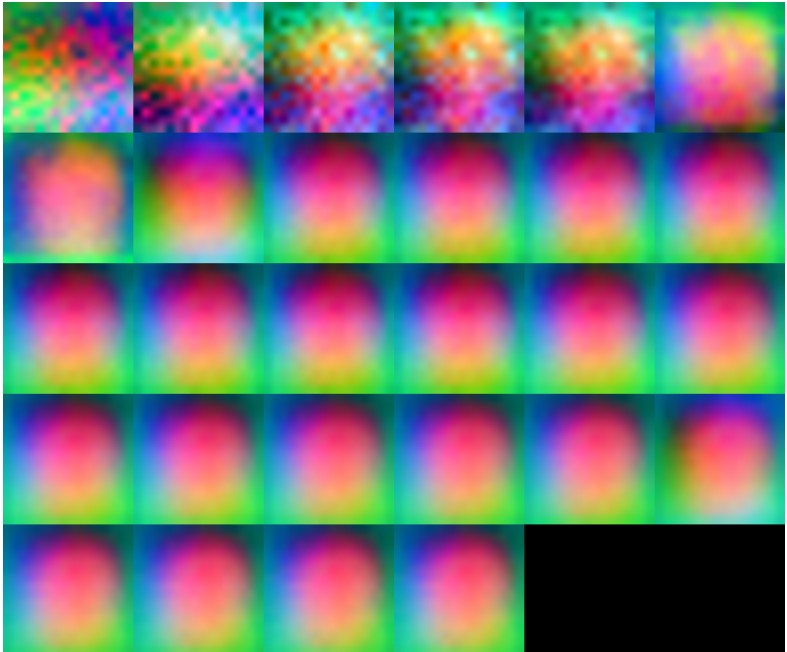

Figure 21: **Visualization of SiT-XL/2 model + LayerSync features with 90% noise added to the input image.** The top-left plot shows the features from the first block, and subsequent blocks are visualized row by row, ending with the final block in the bottom-right corner.

## S   QUALITATIVE EXAMPLES

We provide qualitative examples in Figure 22. The model is trained for 800 epochs on ImageNet dataset (Deng et al., 2009) and the samples are generated using classifier-free guidance with a scale of 4 and the ODE Heun sampler.

Additionally, qualitative comparisons between the baseline SiT-XL/2, SiT-XL/2 regularized with Dispersive, and SiT-XL/2 regularized with LayerSync trained on ImageNet dataset are shown in Figure 23. All models are trained for 400K iterations and share the same noise, sampler, and number of sampling steps. The samples are generated using ODE Heun sampler and no classifier-free guidance is used. LayerSync improves generation quality without relying on external representation.

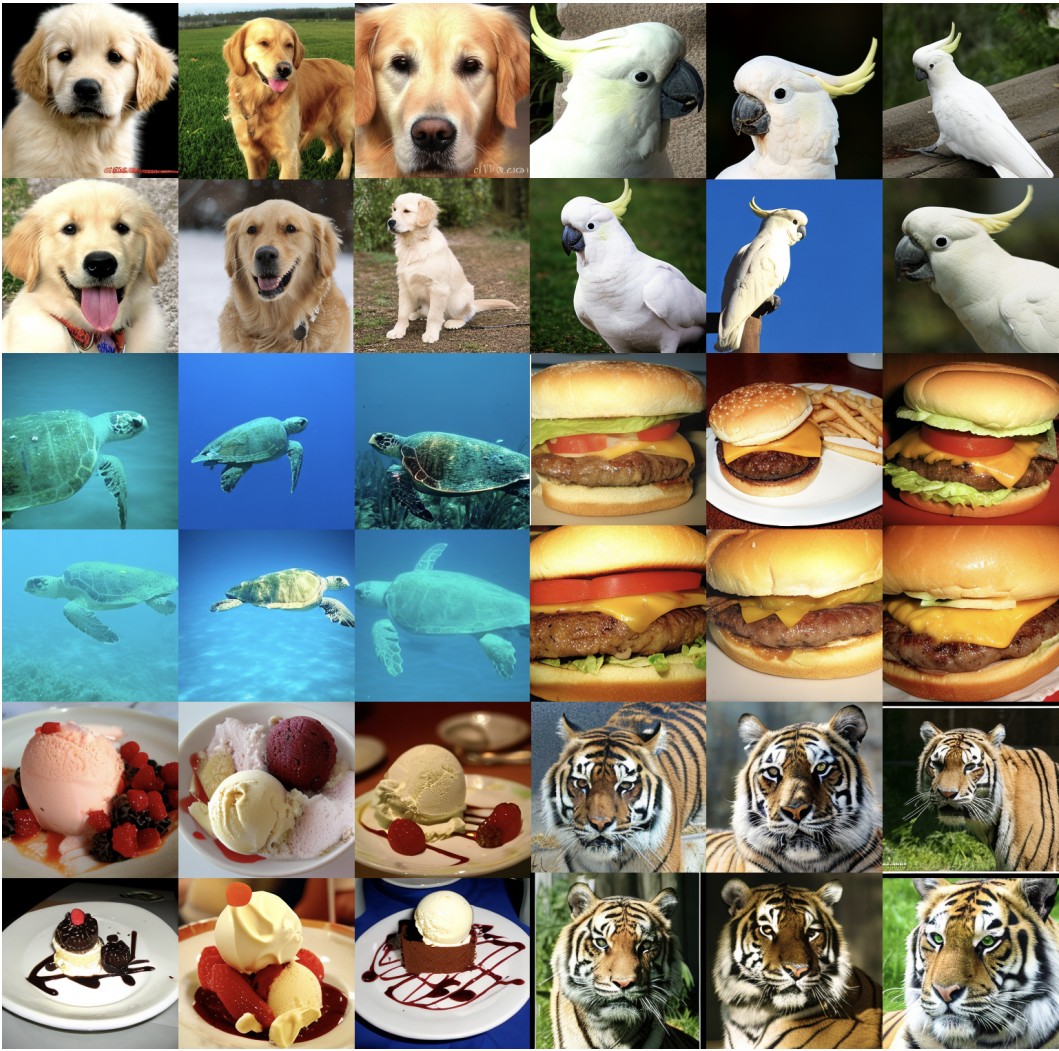

Figure 22: **Selected samples from the SiT XL/2 with LayerSync on ImageNet 256×256.** We use classifier-free guidance with a CFG of 4.0.

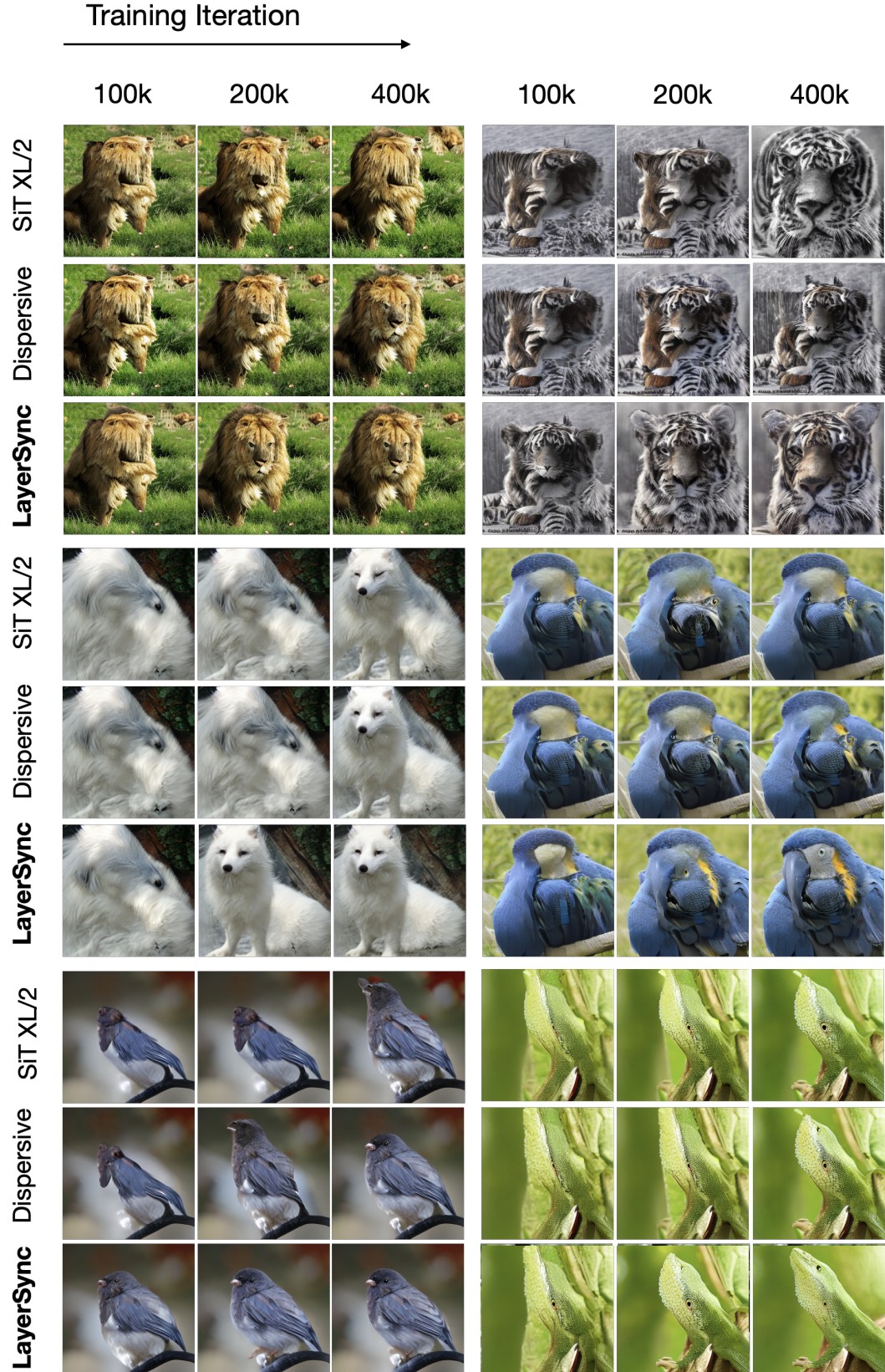

Figure 23: Qualitative comparison of SiT-XL/2 when regularized with Dispersive and LayerSync.

