# OpenReview forum: "LayerSync: Self-aligning Intermediate Layers"
_ICLR.cc/2026/Conference — ICLR 2026 Poster_

### Official Review · Reviewer_5YQE · 2025-10-30

**Soundness:** 4
**Presentation:** 2
**Contribution:** 3
**Rating:** 4
**Confidence:** 4

**Summary:**

The paper presents LayerSync, a self-contained, plug-and-play regularization method for diffusion models leveraging their own intermediate representation. Based on the analysis on internal representations, LayerSync uses the model’s semantically rich layers as intrinsic guidance requiring no additional parameters or external components. Experimental results show consistent improvements in generation quality and training efficiency across multiple domains including, image, audio, video, and human motion.

**Strengths:**

- The overall flow of the paper is clear. To overcome the limitation of relying on external alignment, the authors first analyze the internal representations of diffusion models using discriminative tasks. They then propose a simple self-contained alignment between earlier and later layers, where the latter hold richer semantic information. Finally, they validate the idea through cross-domain experiments and detailed analysis of internal representations.

- The main strengths of the paper are broad generalization and thorough experiments. The proposed method is lightweight and domain-agnostic, showing consistent improvements across different modalities without the need for extra losses or external modules.

**Weaknesses:**

- The paper gives a simple heuristic for choosing which layers to align (Section 3.4), but more experiments on this would help. In Figure 3, layer 20 seems to work best for two tasks, yet the paper aligns layer 8 → 16 instead. Also, both SiT-L and SiT-XL use the same 8th layer (from Table 8). A layer-pair ablation could show which combinations are most effective and why.
- The authors claim that “LayerSync establishes a virtuous cycle that progressively refines the entire feature hierarchy,” but this isn’t clearly supported. It would be more convincing to show how the representations evolve during training or how this behavior changes with different layer selections.
- It seems the alignment is applied without considering timestep. Is there any analysis on how aligning at different timestep affects the results?

**Questions:**

- How does LayerSync perform with CFG in Table 2?
- The overall writing and presentation could be improved for better readability. Some sections, such as Section 4.3 (Results), would need detailed explanations. There are also a few minor typos and formatting issues (e.g., line 265 “table → Table”, 266 “SiTXL → SiT-XL”, 327 “10.000 → 10,000”, 355 missing space after “Results.”, 359 missing comma before “we”, Eq (6) notation issue).

---

> ### Author Response · Authors · 2025-11-22
>
> We thank the reviewer for their time and for the valuable feedback to improve our paper. We will improve the writing and presentation of the paper in the updated version. Below we have provided answers for the raised questions.

---

> ### Author Response · Authors · 2025-11-22
>
> **1. Ablation on Layers.**
>
> We conducted an in-depth ablation study over block selection strategy for different architectures (SiT-L and SiT-XL). We trained all the models for 80 epochs and provided quantitative results in Table 1-6. As discussed in the main paper section 3.4, there is a wide robust region wherein varying the blocks being synced does not vary the results significantly (Table 1 and Table 4). Additionally, when the distance between the blocks is low the gains are suboptimal (Table 2 and Table 5). Finally, aligning with the decoder blocks (very last layers), the gains are lower (Table 3 and Table 6).
>
> LayerSync increases the correlation between the intermediate blocks of the model. We argue that such correlations emerge naturally at convergence in Transformer. In particular, as discussed in [1], the blocks in the ViT architecture are highly correlated and are structured in three groups. The first group focuses on local features, the middle group consists of blocks that are highly correlated with each other and capture global features, and the last group behaves as a decoder [1]. An example of such groups for SiTXL trained on ImageNet dataset at convergence is provided in Link1.
>
> While at convergence the blocks naturally separate into three groups, we observe that this structure does not appear at the beginning of training. An example of vanilla SiTXL after 120k iterations of training is provided in Link2. In LayerSync, we impose, early in training, through an extra loss factor the structure that the model would naturally reach only after long training. An example of SiTXL with LayerSync after 120k iterations of training is provided in Link3.
>
> We argue that as long as the blocks before the third group are aligned, training is accelerated. Implicitly, what LayerSync does is adjusting the relative sizes of the first two groups of blocks. For reference block selection, one important criteria that should be taken into account is that it has been shown that having the early transformer blocks focus on local features is beneficial [3], we should avoid making the first group too small, hence we avoid matching with the very first blocks.
>
> While LayerSync increases the correlation within the second group of blocks, one might worry that this could lead to collapse or redundancy across layers. However, it has already been observed in transformer architectures that the middle blocks in particular are naturally very highly correlated, so this behavior is intrinsic to the model rather than an artifact of our method [2]. Although there have been attempts to prune or remove these blocks, prior work on LLMs shows that, while intermediate blocks can sometimes be removed without significantly affecting performance on easy tasks such as simple question answering, they remain important for more challenging tasks [2]. We observe a similar pattern in image generation: removing the correlated blocks degrades FID, and the model does not recover from this loss in performance.
>
> In general, LayerSync improves the representations of earlier blocks by self-aligning them with layers that already exhibit stronger features, creating a virtuous cycle of representation refinement and additionally imposing a structure between the blocks which will naturally happen at convergence.
>
>
> Link1: https://anonymous.4open.science/api/repo/LayerSync-rebuttal-FBDF/file/Images/sitxl.png?v=fa7fddad
>
> Link2: https://anonymous.4open.science/api/repo/LayerSync-rebuttal-FBDF/file/Images/vanilla_120k.png?v=c40d92f3
>
> Link3:
> https://anonymous.4open.science/api/repo/LayerSync-rebuttal-FBDF/file/Images/8-18-120k.png?v=00794430
>
> Table1:https://anonymous.4open.science/api/repo/LayerSync-rebuttal-FBDF/file/Images/Table1.png?v=4b929f73
>
> Table 2: https://anonymous.4open.science/api/repo/LayerSync-rebuttal-FBDF/file/Images/Table2.png?v=a53a5a27
>
> Table 3: https://anonymous.4open.science/api/repo/LayerSync-rebuttal-FBDF/file/Images/Table3.png?v=8db44c4c
>
> Table 4: https://anonymous.4open.science/api/repo/LayerSync-rebuttal-FBDF/file/Images/Table4.png?v=8f3a06be
>
> Table 5: https://anonymous.4open.science/api/repo/LayerSync-rebuttal-FBDF/file/Images/Table%205.png?v=37631a12
>
> Table 6: https://anonymous.4open.science/api/repo/LayerSync-rebuttal-FBDF/file/Images/Table%206.png?v=1d77f7ca
>
> [1] Raghu, Maithra, et al. "Do vision transformers see like convolutional neural networks?." Advances in neural information processing systems 34 (2021): 12116-12128.
>
> [2] Gromov, Andrey, et al. "The unreasonable ineffectiveness of the deeper layers." arXiv preprint arXiv:2403.17887 (2024).
>
> [3] An, Jie, et al. "On Inductive Biases That Enable Generalization in Diffusion Transformers." The Thirty-ninth Annual Conference on Neural Information Processing Systems. 2024.

---

> ### Author Response · Authors · 2025-11-22
>
> **2. The virtuous cycle and the evolution of representations over time and over different matches.**
>
> We thank the reviewer for this insightful comment. To provide empirical support for the "virtuous cycle" and analyze the evolution of the feature hierarchy, we conducted two additional studies evaluating the segmentation performance (mIoU) of internal representations throughout the network on PASCAL VOC dataset.
>
> First, we tracked the evolution of representations throughout the training process (from 100k to 600k steps). As shown in Figure 1, we observe that while the relative structure of feature quality across layers remains stable, the performance monotonically improves across the entire hierarchy.
>
> Our second study evaluates models trained with different alignment targets at the same training stage (100k steps) (see Figure 2). This comparison provides the most compelling evidence for our hypothesis. When we synchronize an early block (e.g., block 6) with a deeper target (block 18 vs. block 10), we observe two critical effects:
>
> **- Global Improvement:** The model guided by the deeper layer achieves superior downstream performance and lower FID (15.39 vs. 23.90), indicating that guidance from deeper layers correlates with better overall representations.
>
> **- Accelerated Maturation:** Notably, using a deeper target shifts the peak performance of the network to earlier layers. By effectively "pulling" semantic richness from the deep target to the earlier block, the early layers appear to acquire higher-level features sooner in the depth hierarchy.
>
> These observations are consistent with the hypothesized virtuous cycle: guiding an early block with a stronger target improves its representation, which in turn provides higher-quality input to subsequent layers. This likely facilitates the learning of stronger deep representations, which then serve as even better guides, progressively refining the entire hierarchy.
>
>
>
> Figure_1: https://anonymous.4open.science/api/repo/LayerSync-rebuttal-FBDF/file/Images/segmentation.png?v=dbaa06d8
>
> Figure_2:
> https://anonymous.4open.science/api/repo/LayerSync-rebuttal-FBDF/file/Images/segmentation-pascalvoc.png?v=dbe1b903
>
> **3. Alignment considering the timestep.**
>
> We thank the reviewer for this comment. It is indeed interesting to study the effect timestep on alignment. Thus we designed an experiment where we applied alignment only on the last 75 %, 50 % and 25 % of timesteps. We trained SiTXL/2 on ImageNet 256*256 for 80 epochs and applied LayerSync on the mentioned timesteps. Interestingly we observed that the best performance is achieved when alignment is applied on all the timesteps which confirms that improving the weak representations is beneficial regardless of the timestep.
>
> | Timestep   | FID |
> |---------|-------:|
> | 25%  | 18.28  |
> | 50%|  18.77 |
> | 75%  | 17.68 |
> | 100%|  16.03  |
>
> **4. Table 2 with CFG.**
>
> In table 2 all the baselines are already with cfg. We additionally applied guidance approach proposed in [1] to all the methods listed below, we will add these extra results to Table 2 for completeness.
>
> | Method   | FID |
> |---------|-------:|
> | REPA  | 1.42  |
> | LayerSync|  1.50  |
>
> [1] Kynkäänniemi, Tuomas, et al. "Applying guidance in a limited interval improves sample and distribution quality in diffusion models." Advances in Neural Information Processing Systems 37 (2024): 122458-122483.
>
> Finally, we would like to thank the reviewer again for their constructive feedbacks. We are highly responsive during the rebuttal period and warmly welcome any follow-up questions or further discussion.

---

> > ### Comment · Reviewer_5YQE · 2025-11-27
> >
> > Thanks a lot to the authors for additional experiments and analysis during the rebuttal period. Most of my concerns are resolved, thereby raising my score to 6.

---

### Official Review · Reviewer_eqEG · 2025-10-31

**Soundness:** 1
**Presentation:** 2
**Contribution:** 1
**Rating:** 2
**Confidence:** 5

**Summary:**

The paper presents LayerSync as a self-sufficient, plug-and-play regularization for diffusion models across modalities, but the claims are undermined by a lack of solid theoretical grounding and insufficiently rigorous empirical validation. The experimental comparisons appear incomplete, and I identified errors in data analysis that could misrepresent the method’s effectiveness (e.g., the reported speedups and generalization results may be overstated). Without stronger theoretical justification, more comprehensive baselines, and corrected analyses, the work risks misleading reviewers and does not convincingly support its central contributions.

**Strengths:**

- The core idea behind LayerSync is notably simple and clear, which contributes to the overall readability and accessibility of the paper.
- The authors provide comprehensive empirical validation across multiple domains, including image, video, and audio generation, demonstrating the versatility of their approach.
- Compared to the selected baselines, the proposed method exhibits a tangible improvement in convergence speed, as evidenced by the experimental results.

**Weaknesses:**

- The choice of SiT as the sole baseline is not sufficiently convincing. Although Table 1 demonstrates some improvements when LayerSync is added to SiT under various model sizes, the baseline itself performs relatively poorly. Achieving gains over a weak baseline does not provide strong evidence of the proposed method's effectiveness.
- In the experiments presented in Table 2, LayerSync does not outperform SiT-XL/x+REPA or SiT-XL+REED in terms of FID or training epochs. However, the authors highlight their own results in bold, which may mislead reviewers. Furthermore, the experimental comparison remains limited to relatively weak baselines and does not include more recent and competitive approaches such as DJEPA or VAR. As a result, the evaluation is not comprehensive, and it is difficult to assess the true effectiveness of the proposed architecture.。

**Questions:**

- The paper lacks solid theoretical analysis of the LayerSync mechanism itself. In the original SiT architecture, skip connections between blocks already help maintain feature consistency across layers. The additional loss introduced by LayerSync may simply increase the overall gradient magnitude, which could accelerate convergence in a manner similar to using a larger learning rate. While this is a hypothesis, it raises questions about the underlying mechanism driving the observed improvements.
- I strongly encourage the authors to provide more theoretical insights and empirical evidence regarding LayerSync in their rebuttal. For example, is the gradient norm significantly larger compared to the baseline? Would directly amplifying the feature gradients at corresponding locations in the baseline yield similar training benefits? Addressing these points would greatly clarify the contribution and effectiveness of LayerSync.

---

> ### Author Response · Authors · 2025-11-21
>
> We thank the reviewer for their time and for the valuable feedback to improve our paper. We acknowledge your concerns and address them below. We would be happy to engage in further discussion and clarify any remaining questions.
>
> **Concern1: LayerSync effect is similar to increasing the learning rate**
>
> We want to thank the reviewer for raising this concern. It is indeed important to show that LayerSync impact is not simply due to an increase in the gradient magnitude. Therefore, we
> designed two sets of experiments below. All the models are trained with 16 GPUs, batch size 1024 for 100k iterations (80 epoches). We then report the gradient norm and the FID of different configurations. We consistently observed that **the gradient norm with LayerSync is actually smaller than that of the baseline**, indicating that our method is not simply equivalent to using a larger learning rate.
>
> (1) Global learning rate increase.
>
> Starting from the default learning rate of 1×10−4, we trained models with higher learning rates 2×10−4, 5×10−4. For learning rates above 5×10−4, the model diverges. While increasing the learning rate can partially accelerate training, the resulting FID scores remain worse than those obtained with LayerSync with the default learning rate 1×10−4. The visualization of gradient norm is provided here: https://anonymous.4open.science/api/repo/LayerSync-rebuttal-FBDF/file/Images/grad_norm_plot.png?v=021ad90b
> | Method   | lr. | FID |
> |---------|-------------:|-------:|
> | SiTXL/2  | 1e-4 | 26.53  |
> | SiTXL/2  | 2e-4 | 24.95  |
> | LayerSync| 1e-4 |  16.03  |
>
> (2) Higher learning rate on early blocks only.
>
> We then increased the learning rate only for the first 8 blocks and compared this to a model trained with LayerSync aligning layers 8–16 at a global learning rate of 1×10−4. Again, for learning rates above 1×10−3 training diverges. For 2×10−4, 5×10−4, and 1×10−3, the FID improvements do not match those obtained with LayerSync. The visualization of gradient norm is provided here: https://anonymous.4open.science/api/repo/LayerSync-rebuttal-FBDF/file/Images/early_grad_norm_plot.png?v=476012ed
> | Method   | lr. Early Layers | General lr. | FID |
> |---------|-------------:|------------:|-------:|
> | SiTXL/2  | 1e-4 |1e-4 | 26.53    |
> | SiTXL/2  | 2e-4 |1e-4 | 19.24   |
> | SiTXL/2  | 5e-4 |1e-4 | 24.63    |
> | LayerSync| 1e-4 | 1e-4 | 16.03    |
>
> These findings support our claim that LayerSync’s gains arise from improved layer-wise synchronization and representation learning, rather than from implicitly increasing the learning rate or gradient magnitude.

---

> ### Author Response · Authors · 2025-11-21
>
> **Concern 2: Underlying mechanism between the correlated blocks**
>
> We conducted an in-depth ablation study over block selection strategy for different architectures (SiT-L and SiT-XL). We trained all the models for 80 epochs and provided quantitative results in Table 1-6. As discussed in the main paper section 3.4, there is a wide robust region wherein varying the blocks being synced does not vary the results significantly (Table 1 and Table 4). Additionally, when the distance between the blocks is low the gains are suboptimal (Table 2 and Table 5). Finally, aligning with the decoder blocks (very last layers), the gains are lower (Table 3 and Table 6).
>
> LayerSync increases the correlation between the intermediate blocks of the model. We argue that such correlations emerge naturally at convergence in Transformer. In particular, as discussed in [1], the blocks in the ViT architecture are highly correlated and are structured in three groups. The first group focuses on local features, the middle group consists of blocks that are highly correlated with each other and capture global features, and the last group behaves as a decoder [1]. An example of such groups for SiTXL trained on ImageNet dataset at convergence is provided in Link1.
>
> While at convergence the blocks naturally separate into three groups, we observe that this structure does not appear at the beginning of training. An example of vanilla SiTXL after 120k iterations of training is provided in Link2. In LayerSync, we impose, early in training, through an extra loss factor the structure that the model would naturally reach only after long training. An example of SiTXL with LayerSync after 120k iterations of training is provided in Link3.
>
> We argue that as long as the blocks before the third group are aligned, training is accelerated. Implicitly, what LayerSync does is adjusting the relative sizes of the first two groups of blocks. For reference block selection, one important criteria that should be taken into account is that it has been shown that having the early transformer blocks focus on local features is beneficial [3], we should avoid making the first group too small, hence we avoid matching with the very first blocks.
>
> While LayerSync increases the correlation within the second group of blocks, one might worry that this could lead to collapse or redundancy across layers. However, it has already been observed in transformer architectures that the middle blocks in particular are naturally very highly correlated, so this behavior is intrinsic to the model rather than an artifact of our method [2]. Although there have been attempts to prune or remove these blocks, prior work on LLMs shows that, while intermediate blocks can sometimes be removed without significantly affecting performance on easy tasks such as simple question answering, they remain important for more challenging tasks [2]. We observe a similar pattern in image generation: removing the correlated blocks degrades FID, and the model does not recover from this loss in performance.
>
> In general, LayerSync improves the representations of earlier blocks by self-aligning them with layers that already exhibit stronger features, creating a virtuous cycle of representation refinement and additionally imposing a structure between the blocks which will naturally happen at convergence.
>
>
> Link1: https://anonymous.4open.science/api/repo/LayerSync-rebuttal-FBDF/file/Images/sitxl.png?v=fa7fddad
>
> Link2: https://anonymous.4open.science/api/repo/LayerSync-rebuttal-FBDF/file/Images/vanilla_120k.png?v=c40d92f3
>
> Link3:
> https://anonymous.4open.science/api/repo/LayerSync-rebuttal-FBDF/file/Images/8-18-120k.png?v=00794430
>
> Table1:https://anonymous.4open.science/api/repo/LayerSync-rebuttal-FBDF/file/Images/Table1.png?v=4b929f73
>
> Table 2: https://anonymous.4open.science/api/repo/LayerSync-rebuttal-FBDF/file/Images/Table2.png?v=a53a5a27
>
> Table 3: https://anonymous.4open.science/api/repo/LayerSync-rebuttal-FBDF/file/Images/Table3.png?v=8db44c4c
>
> Table 4: https://anonymous.4open.science/api/repo/LayerSync-rebuttal-FBDF/file/Images/Table4.png?v=8f3a06be
>
> Table 5: https://anonymous.4open.science/api/repo/LayerSync-rebuttal-FBDF/file/Images/Table%205.png?v=37631a12
>
> Table 6: https://anonymous.4open.science/api/repo/LayerSync-rebuttal-FBDF/file/Images/Table%206.png?v=1d77f7ca
>
> [1] Raghu, Maithra, et al. "Do vision transformers see like convolutional neural networks?." Advances in neural information processing systems 34 (2021): 12116-12128.
>
> [2] Gromov, Andrey, et al. "The unreasonable ineffectiveness of the deeper layers." arXiv preprint arXiv:2403.17887 (2024).
>
> [3] An, Jie, et al. "On Inductive Biases That Enable Generalization in Diffusion Transformers." The Thirty-ninth Annual Conference on Neural Information Processing Systems. 2024.

---

> > ### Author Response · Authors · 2025-11-21
> >
> > **Concern3: Choice of SiT as a baseline**
> >
> > We thank the reviewer for raising this point. We would like to clarify that the primary goal of our work is not to propose a new architecture to improve image generation quality, but rather to introduce a self-contained, domain-agnostic method to accelerate the training of Diffusion Transformers. Diffusion Transformers have become the de facto architecture for diffusion-based generative models and are now widely used across multiple modalities, including images, videos, audio, and motion. SiT is essentially a Diffusion Transformer trained with flow matching, and our method is designed to plug into this standard architecture without modifying its core design. SiT is the baseline used in all the previous work focused on accelerating the diffusion transformer training.
> >
> > In contrast, the baselines mentioned by the reviewer focus on alternative architectures for autoregressive image generation, typically evaluated on ImageNet, and target a different design space than Diffusion Transformers. While these works are valuable, they lie somewhat outside the main scope of our paper, which is centered on accelerating training within the widely adopted Diffusion Transformer framework rather than replacing it.
> >
> > That said, we agree that including these baselines can help provide a more comprehensive picture. When we apply the guidance approach proposed [1], both REPA and LayerSync outperform the mentioned baselines.
> >
> > | Method   | #params | FID |
> > |---------|-------------:|-------:|
> > | VAR-d20  | 600M | 2.57  |
> > | D-JEPA-L  | 687M | 1.58  |
> > | REPA  | 675M | 1.42  |
> > | LayerSync| 675M |  1.50  |
> >
> > We will add the requested methods to Table 2 of the main paper and briefly discuss their relation to our approach in the revised version, while emphasizing that our contribution is somewhat orthogonal: LayerSync can be seen as a general training-speedup mechanism for Diffusion Transformers rather than a competing architecture.
> >
> > [1] Kynkäänniemi, Tuomas, et al. "Applying guidance in a limited interval improves sample and distribution quality in diffusion models." Advances in Neural Information Processing Systems 37 (2024): 122458-122483.
> >
> > **Concern4: Results shown in bold in Table 2**
> >
> > We thank the reviewer for pointing this out. In that table, our goal was to highlight that we outperform the previous self-contained baseline (Dispersive), while the best overall result (REED) is already shown in bold. To avoid confusion, we will underline our results in the revised version to clearly distinguish them from the bolded state-of-the-art numbers.
> >
> >
> > Finally, we would like to thank the reviewer again for their constructive feedbacks. We are highly responsive during the rebuttal period and warmly welcome any follow-up questions or further discussion.

---

### Official Review · Reviewer_2qbB · 2025-10-31

**Soundness:** 3
**Presentation:** 3
**Contribution:** 3
**Rating:** 6
**Confidence:** 4

**Summary:**

This paper aims to accelerate the training of diffusion transformers without external guidance to avoid dependency on an external model that introduces additional training cost and limited domains. To this end, the authors propose LayerSync, a simple yet effective model that uses features from a later layer of the diffusion model itself as an alignment target. They show that LayerSync accelerates training of diffusion models in a domain-agnostic manner.

**Strengths:**

1. This paper proposes a simple yet effective framework that accelerates the training of the diffusion transformer.

2. This paper shows that LayerSync is modality-agnostic (image, audio, human motion), demonstrating the effectiveness of the proposed method in various domains that may have challenges in using pre-trained representations.

3. This paper introduces the heuristic guidance to find a layer to align, which is highly practical for using diffusion models in different domains.

**Weaknesses:**

1. A comparison to a more alignment method is needed. For instance, despite SRA [1] using the EMA model (which can make training of DiT slower), I think comparison with it in terms of training time (e.g., wall-time clock) is important to claim that we indeed need an alignment strategy that does not depend on an additional module.

2. The authors claim that REPA has limitations in that a pre-trained representation model is not available for various domains. However, several studies have proposed methods to obtain a pre-trained representation model in a domain-agnostic manner [2-4]. If we use such models, we may accelerate the training of the diffusion model even faster than LayerSync (in terms of training time), and we can use pre-computed features to reduce the memory cost.

[1] Jiang et al., No Other Representation Component Is Needed: Diffusion Transformers Can Provide Representation Guidance by Themselves, Arxiv 2025 \
[2] Tamkin et al., DABS: A Domain-Agnostic Benchmark for Self-Supervised Learning, NeurIPS 2021 \
[3] Baevski et al., Efficient Self-supervised Learning with Contextualized Target Representations for Vision, Speech and Language, Arxiv 2022 \
[4] Jang et al., Modality-Agnostic Self-Supervised Learning with Meta-Learned Masked Auto-Encoder, NeurIPS 2023

**Questions:**

Please answer the weaknesses.

---

> ### Author Response · Authors · 2025-11-22
>
> We thank the reviewer for their time and for the valuable feedback to improve our paper. Below we have provided answers for the raised questions.
>
> **1. Comparison with SRA.**
> We want to thank the reviewer for mentioning SRA baseline. Indeed it is critical to show that LayerSync is effective without requiring any EMA model or external models. Thus, we compare the FID of LayerSync and SRA, for the sake of fairness both models are trained for 800 epochs and the guidance approach proposed in [1] is applied on both. We observe that LayerSync outperforms SRA which further showcases the effectiveness of LayerSync. We will add these results to the Table 2 of the main paper.
>
> | Method   | FID |
> |---------|-------:|
> | REPA  | 1.42  |
> | SRA  | 1.58  |
> | LayerSync|  1.50  |
>
> We will additionally provide a wall-time clock comparison between the approaches as requested by the reviewer.
>
> [1] Kynkäänniemi, Tuomas, et al. "Applying guidance in a limited interval improves sample and distribution quality in diffusion models." Advances in Neural Information Processing Systems 37 (2024): 122458-122483.
>
> **2. Availability of external representations.**
>
> We agree with the reviewer that it is possible to obtain a new pre-trained model for each domain if needed. And it is likely that such approach would accelerate training of diffusion model even more. However, this method is costly and we propose an approach to avoid that. Furthermore, in our rebuttal experiments, we found that combining LayerSync with REPA yielded better performance than either method alone, suggesting that the internal structural alignment of LayerSync (see Comment 1 for reviewer Sohi) and the external semantic injection of REPA are complementary axes of improvement. Additionally, we observe that applying REPA to a layer between two syncing layers is more effective than applying it before the syncing layers. The results summarized in the table below are after 40 epochs of training, we continue training the models and provide further updates during the rebuttal time.
>
> | Method   | REPA Layer | LayerSync Layer | FID |
> |---------|:-------------:|:------------:|-------:|
> | SiTXL/2  | - | - | 59.45    |
> | SiTXL/2 + LayerSync  | - | 8-16 | 46.26   |
> | SiTXL/2 + REPA  | 7 |- | 46.06    |
> | SiTXL/2 + REPA + LayerSync  | 7 |8-16 | 43.55    |
> | SiTXL/2 + REPA + LayerSync| 10| 8-16 | 29.68    |
>
>
> Finally, we would like to thank the reviewer again for their constructive feedback. We are highly responsive during the rebuttal period and warmly welcome any follow-up questions or further discussion.

---

> > ### Author Response · Authors · 2025-12-01
> >
> > We want to thank the reviewer for their time and valuable feedback. We provide below a more detailed comparison between SRA [1] and our work LayerSync.
> >
> > We compare the two approaches in terms of training time and computational overhead. We computed the metrics for SiT-XL/2 using 4 GH200 GPUs and a batch size of 32 per GPU. Results are reported in the table below. We show that LayerSync requires 25.5\% fewer Flops, is 40.5\% faster in real-time, and reaches an FID 5\% lower.
> >
> > | Method | FID ↓ | Wall-clock time/step ↓ | GFlops ↓ |
> > |:------------------------:|:--------:|:----------------------:|:----------:|
> > | SiT-XL/2 + SRA | 1.58 | 0.617 | 30,762 |
> > | **SiT-XL/2 + LayerSync** | **1.50** | **0.367** | **22,910** |
> >
> > [1] Jiang et al., No Other Representation Component Is Needed: Diffusion Transformers Can Provide Representation Guidance by Themselves, Arxiv 2025

---

### Official Review · Reviewer_Sohi · 2025-11-03

**Soundness:** 3
**Presentation:** 3
**Contribution:** 2
**Rating:** 6
**Confidence:** 5

**Summary:**

This paper introduces LayerSync, a parameter-free, self-contained regularization strategy for diffusion models. Unlike previous work that aligns diffusion model representations with large external models, LayerSync aligns a model’s own weaker intermediate layers to its stronger, semantically richer layers, thus providing internal guidance. The method is simple (involving an intra-model alignment loss over patchwise cosine similarities), incurs negligible overhead, and is broadly applicable across data modalities—including images, audio, motion, and video. Empirical results on ImageNet, MTG-Jamendo (audio), HumanML3D (motion), and CLEVRER (video), as well as comprehensive ablations and internal feature analysis, demonstrate that LayerSync consistently outperforms prior self-contained baselines and narrows the performance gap with externally guided methods.

**Strengths:**

- LayerSync is evaluated across four distinct modalities (image, audio, motion, and video), each with strong quantitative results and improvements in training efficiency, generation quality, and representation quality (Tables 1–4, 7; Figure 1). These cross-domain demonstrations suggest the approach’s generality.
- Unlike methods requiring external pretrained models (e.g., DINOv2, VLMs), LayerSync requires no additional data or parameters, making it attractive for non-visual domains and resource-limited scenarios.
- For image generation on ImageNet, LayerSync accelerates convergence by over 8.75x (Table 1/Figure 1b) compared to baseline diffusion transformers, and outpaces self-contained alternatives like Dispersive.
- Qualitative and quantitative comparisons (e.g., Figure 2, Figure 3, Figure 15-16) support claims about quality and internal changes, and supplementary figures show robust improvements (e.g., noise robustness, block dropping effects).

**Weaknesses:**

- The strategy for picking “strong” and “weak” layers is based on architectural distance and exclusion of “final 20%” blocks. While Table 5 suggests robustness to randomization, the method remains ad-hoc, and a more systematic (possibly data-driven or information-theoretic) approach could yield better guarantees. There’s no analysis as to how the reference layer selection interacts with model depth or varying data/architecture.
- The paper is primarily empirical; while this is fine, and the practical results are compelling, the absence of theoretical grounding for the expected improvements (e.g., analysis on generalization, capacity, information propagation) or limitations of the approach means its scope is less clear. For instance, are there pathological settings (e.g., excessive over-alignment leading to collapse) or fundamental limits with this approach where external representations would still be needed?
- Although the paper notes that training with LayerSync increases resilience to block removal, it is not clear whether stronger self-alignment could lead to diminished feature diversity, especially in representations, and whether downstream performance is robust to such collapse.
- For audio, motion, and video, the paper mostly reports aggregate scores (e.g., FAD, FVD, FID) with limited visualization or error analysis of failure modes. Given the modality-general claim, richer qualitative evidence would reinforce the universality claim.

**Questions:**

Please refer to Weaknesses

---

> ### Author Response · Authors · 2025-11-21
>
> We want to thank the reviewer for their time and constructive feedback. We have provided a link to qualitative results in the general comment. Below, we address all the raised questions.

---

> ### Author Response · Authors · 2025-11-21
>
> **1. Layer Selection Strategy.**
> We conducted an in-depth ablation study over block selection strategy for different architectures (SiT-L and SiT-XL). We trained all the models for 80 epochs and provided quantitative results in Table 1-6. As discussed in the main paper section 3.4, there is a wide robust region wherein varying the blocks being synced does not vary the results significantly (Table 1 and Table 4). Additionally, when the distance between the blocks is low the gains are suboptimal (Table 2 and Table 5). Finally, aligning with the decoder blocks (very last layers), the gains are lower (Table 3 and Table 6).
>
> LayerSync increases the correlation between the intermediate blocks of the model. We argue that such correlations emerge naturally at convergence in Transformer. In particular, as discussed in [1], the blocks in the ViT architecture are highly correlated and are structured in three groups. The first group focuses on local features, the middle group consists of blocks that are highly correlated with each other and capture global features, and the last group behaves as a decoder [1]. An example of such groups for SiTXL trained on ImageNet dataset at convergence is provided in Link1.
>
> While at convergence the blocks naturally separate into three groups, we observe that this structure does not appear at the beginning of training. An example of vanilla SiTXL after 120k iterations of training is provided in Link2. In LayerSync, we impose, early in training, through an extra loss factor the structure that the model would naturally reach only after long training. An example of SiTXL with LayerSync after 120k iterations of training is provided in Link3.
>
> We argue that as long as the blocks before the third group are aligned, training is accelerated. Implicitly, what LayerSync does is adjusting the relative sizes of the first two groups of blocks. For reference block selection, one important criteria that should be taken into account is that it has been shown that having the early transformer blocks focus on local features is beneficial [3], we should avoid making the first group too small, hence we avoid matching with the very first blocks.
>
> While LayerSync increases the correlation within the second group of blocks, one might worry that this could lead to collapse or redundancy across layers. However, it has already been observed in transformer architectures that the middle blocks in particular are naturally very highly correlated, so this behavior is intrinsic to the model rather than an artifact of our method [2]. Although there have been attempts to prune or remove these blocks, prior work on LLMs shows that, while intermediate blocks can sometimes be removed without significantly affecting performance on easy tasks such as simple question answering, they remain important for more challenging tasks [2]. We observe a similar pattern in image generation: removing the correlated blocks degrades FID, and the model does not recover from this loss in performance.
>
> In general, LayerSync improves the representations of earlier blocks by self-aligning them with layers that already exhibit stronger features, creating a virtuous cycle of representation refinement and additionally imposing a structure between the blocks which will naturally happen at convergence.
>
>
> Link1: https://anonymous.4open.science/api/repo/LayerSync-rebuttal-FBDF/file/Images/sitxl.png?v=fa7fddad
>
> Link2: https://anonymous.4open.science/api/repo/LayerSync-rebuttal-FBDF/file/Images/vanilla_120k.png?v=c40d92f3
>
> Link3:
> https://anonymous.4open.science/api/repo/LayerSync-rebuttal-FBDF/file/Images/8-18-120k.png?v=00794430
>
> Table1:https://anonymous.4open.science/api/repo/LayerSync-rebuttal-FBDF/file/Images/Table1.png?v=4b929f73
>
> Table 2: https://anonymous.4open.science/api/repo/LayerSync-rebuttal-FBDF/file/Images/Table2.png?v=a53a5a27
>
> Table 3: https://anonymous.4open.science/api/repo/LayerSync-rebuttal-FBDF/file/Images/Table3.png?v=8db44c4c
>
> Table 4: https://anonymous.4open.science/api/repo/LayerSync-rebuttal-FBDF/file/Images/Table4.png?v=8f3a06be
>
> Table 5: https://anonymous.4open.science/api/repo/LayerSync-rebuttal-FBDF/file/Images/Table%205.png?v=37631a12
>
> Table 6: https://anonymous.4open.science/api/repo/LayerSync-rebuttal-FBDF/file/Images/Table%206.png?v=1d77f7ca
>
> [1] Raghu, Maithra, et al. "Do vision transformers see like convolutional neural networks?." Advances in neural information processing systems 34 (2021): 12116-12128.
>
> [2] Gromov, Andrey, et al. "The unreasonable ineffectiveness of the deeper layers." arXiv preprint arXiv:2403.17887 (2024).
>
> [3] An, Jie, et al. "On Inductive Biases That Enable Generalization in Diffusion Transformers." The Thirty-ninth Annual Conference on Neural Information Processing Systems. 2024.

---

> ### Author Response · Authors · 2025-11-22
>
> **2. Effect of strong self-alignment.**
> Similarly to any regularizer, an over-regularization would degrade the generation quality. However as shown in Table 6. of the main paper, the result is not overly sensitive to variation in this hyperparameter scale.
>
> As shown in REPA [1], when the generation quality degrades, the features representations deteriorate accordingly. Since LayerSync significantly improves FID, it indicates that in our reported operating region, the method is not collapsing features. Even more so, in our representation study in section 4.4, we remark a higher performance on downstream tasks for a LayerSynced model than the vanilla baseline with the very same FID. Additionally, the correlation that LayerSync imposes primarily reduces the angular distance between layers rather than their scale. Thus, high angular similarity does not imply that the features are exactly identical.
>
> [1] Yu, Sihyun, et al. "Representation alignment for generation: Training diffusion transformers is easier than you think." arXiv preprint arXiv:2410.06940 (2024).
>
> **3. Limitations and links to External Representations Methods.**
> The reviewer asks if there are limits where external representations are still needed. In fact, far from replacing one another, we found the two approaches are synergistic. In our rebuttal experiments, combining LayerSync with REPA yielded better performance than either method alone, suggesting that the internal structural alignment of LayerSync and the external semantic injection of REPA are complementary axes of improvement. Additionally, we observe that applying REPA to a layer between two syncing layers is more effective than applying it before the syncing layers. The results summarized in the table below are after 40 epochs of training, we continue training the models and provide further updates during the rebuttal time.
>
> | Method   | REPA Layer | Synced Layer | FID |
> |---------|:-------------:|:------------:|-------:|
> | SiTXL/2  | - | - | 59.45    |
> | SiTXL/2 + LayerSync  | - | 8-16 | 46.26  |
> | SiTXL/2 + REPA  | 7 |- | 46.06 |
> | SiTXL/2 + REPA + LayerSync  | 7 |8-16      | 43.55    |
> | SiTXL/2 + REPA + LayerSync| 10| 8-16      | 29.68    |
>
> **4. Richer Qualitative Evidence.**
> We agree with your recommendation and provide qualitative evidence in the link in the general comment.
>
> Finally, we would like to thank the reviewer again for their constructive feedback. We are highly responsive during the rebuttal period and warmly welcome any follow-up questions or further discussion.

---

### Author Response · Authors · 2025-11-21

We thank all the reviewers for their time, their support of our work, and their valuable feedback to help improve the paper. We have prepared individual responses to address each reviewer’s specific points. We welcome any follow-up discussion and are happy to clarify remaining questions.
Additionally, we would like to expand upon two general points that we believe are of interest to all reviewers:

**1. Improved Quantitative Results with Interval-Based CFG.** During the rebuttal period, we extended our quantitative evaluation by applying Classifier-Free Guidance (CFG) within a limited interval [1], a method similarly explored in both REPA [2] and SRA [3]. Using this sampling approach,  We improved our best FID to 1.50 (improved from our previous best of 1.89). The results are summarized in the table below. Note that all methods listed use the same sampling approach [1] for a fair comparison:
| Method   | FID |
|---------|-------:|
| REPA  | 1.42  |
| SRA  | 1.58  |
| LayerSync|  1.50  |

**2. Additional Qualitative Results.** As requested by the reviewers, we have provided additional qualitative samples at the following link: https://anonymous.4open.science/api/repo/LayerSync-rebuttal-FBDF/file/LayerSync.io/index.html?v=ffd50aae and in the supplementary material.

We plan to update the final version of the paper to incorporate the additional results and points discussed during the rebuttal.

[1] Kynkäänniemi, Tuomas, et al. "Applying guidance in a limited interval improves sample and distribution quality in diffusion models." Advances in Neural Information Processing Systems 37 (2024): 122458-122483.

[2] Yu, Sihyun, et al. "Representation alignment for generation: Training diffusion transformers is easier than you think." arXiv preprint arXiv:2410.06940 (2024).

[3] Jiang, Dengyang, et al. "No Other Representation Component Is Needed: Diffusion Transformers Can Provide Representation Guidance by Themselves." arXiv preprint arXiv:2505.02831 (2025).

---

> ### Author Response · Authors · 2025-12-01
>
> Dear AC,
>
> We sincerely appreciate the time and effort you dedicated to reviewing our manuscript. Thanks to the constructive feedback received during the rebuttal, we have conducted additional experiments which are now included in the revised main paper and appendix. While detailed responses to each reviewer can be found below, we provide here a summary of the main improvements achieved during the rebuttal period:
> - We have grounded our layer selection strategy in previous works [1, 2, 3] that study the internal structure of Diffusion Transformers at convergence, and we validated this approach with extensive ablation studies.
> - We demonstrated that LayerSync is fundamentally different from optimization heuristics such as simply increasing the learning rate.
> - We improved our best **FID on ImageNet to 1.50**.
> - We added a comparison to SRA [4] regarding wall-clock time, FLOPs, and FID. LayerSync outperforms SRA’s reported **FID by 5%** while requiring **60% less wall-clock** time and **30% fewer FLOPs**.
> - We established that **LayerSync is complementary to methods relying on external guidance**, such as REPA [5], and can be combined with these approaches to further accelerate training.
>
> The revised manuscript includes these new studies, analyses, and comparative benchmarks.
>
> Once more, we thank all the reviewers sincerely for their time, their support of our work, and their valuable feedback that helped improve the paper.
>
>
> [1] Raghu, Maithra, et al. "Do vision transformers see like convolutional neural networks?." Advances in neural information processing systems 34 (2021): 12116-12128.
>
> [2] Gromov, Andrey, et al. "The unreasonable ineffectiveness of the deeper layers." arXiv preprint arXiv:2403.17887 (2024).
>
> [3] An, Jie, et al. "On Inductive Biases That Enable Generalization in Diffusion Transformers." The Thirty-ninth Annual Conference on Neural Information Processing Systems. 2024.
>
> [4] Jiang, Dengyang, et al. "No Other Representation Component Is Needed: Diffusion Transformers Can Provide Representation Guidance by Themselves." arXiv preprint arXiv:2505.02831 (2025).
>
> [5] Yu, Sihyun, et al. "Representation alignment for generation: Training diffusion transformers is easier than you think." arXiv preprint arXiv:2410.06940 (2024).

---

### Meta-Review · Area_Chair_TimN · 2026-01-07

**Summary:**

This paper presents a method for improving diffusion model training by regularizing with the transformers' own intermediate representations.
The submission received mixed reviews from the reviewers.
The reviewers mainly recognize the simplicity of the approach, efficiency and practicality, and generalization to broad domains of various modalities.
The main concerns from the reviewers were the lack of theoretical depth (Sohi and eqEG), unclear layer selection strategies (Sohi and 5YQE), and limited interpretability (Sohi and 5YQE). Additionally, 2qbB is concerned about the necessity of the proposed method compared to existing methods, and eqEG is concerned about limited evaluations against stronger baselines as well as the presentation of results.
After reading the paper, the reviewers' comments and the authors' rebuttal, the AC believes the authors' responses would have addressed the reviewers' major concerns. While the issues raised by eqEG are valid, they do not provide strong enough grounds for rejection. The AC believes the merits from the paper outweighs the weaknesses and thus recommends acceptance.

**Reviewer Concerns:**

Reviewers' concerns mostly addressed:
- Layer selection strategies (Sohi, 5YQE)
- More complete comparisons against SRA (2qbB, eqEG)
- Gradient magnitude and learning rate (eqEG)
- Timestep sensitivity (5YQE)

Outstanding concerns:
- Theoretical grounding (Sohi, eqEG)
- Error analysis on failure modes (Sohi)

**Reviewer Scores:**

I think eqEG would raise their rating from 2 to 4, while the rest would keep their ratings of 6.

---

### Decision · Program_Chairs · 2026-01-26

Accept (Poster)